# ABHD4-dependent developmental anoikis safeguards the embryonic brain

Zsófia I. László [1,2,8], Zsolt Lele [1,8], Miklós Zöldi[1,2], Vivien Miczán [1,3], Fruzsina Mógor[1], Gabriel M. Simon[4], Ken Mackie [5], Imre Kacskovics [6,7], Benjamin F. Cravatt [4] & István Katona [1,5 ✉]

A specialized neurogenic niche along the ventricles accumulates millions of progenitor cells in the developing brain. After mitosis, fate-committed daughter cells delaminate from this germinative zone. Considering the high number of cell divisions and delaminations taking place during embryonic development, brain malformations caused by ectopic proliferation of misplaced progenitor cells are relatively rare. Here, we report that a process we term developmental anoikis distinguishes the pathological detachment of progenitor cells from the normal delamination of daughter neuroblasts in the developing mouse neocortex. We identify the endocannabinoid-metabolizing enzyme abhydrolase domain containing 4 (ABHD4) as an essential mediator for the elimination of pathologically detached cells. Consequently, rapid ABHD4 downregulation is necessary for delaminated daughter neuroblasts to escape from anoikis. Moreover, ABHD4 is required for fetal alcohol-induced apoptosis, but not for the well-established form of developmentally controlled programmed cell death. These results suggest that ABHD4-mediated developmental anoikis specifically protects the embryonic brain from the consequences of sporadic delamination errors and teratogenic insults.

[1] Momentum Laboratory of Molecular Neurobiology, Institute of Experimental Medicine, 1450 Budapest Pf. 67., Budapest, Hungary. [2] School of Ph.D. Studies, Semmelweis University, Budapest, Hungary. [3] Faculty of Information Technology and Bionics, Pázmány Péter Catholic University, Budapest, Hungary. [4] The Skaggs Institute for Chemical Biology, Department of Chemical Physiology, The Scripps Research Institute, La Jolla, CA 92307, USA. [5] Department of Psychological and Brain Sciences, Indiana University, Bloomington, IN 47405, USA. [6] Department of Immunology, Eötvös Loránd University, Pázmány Péter stny 1/A., 1117 Budapest, Hungary. [7] ImmunoGenes Ltd, Makkosi út 86., 2092 Budakeszi, Hungary. [8] These authors contributed equally: Zsófia I. László, Zsolt Lele. ✉email: katona@koki.hu

In the developing brain, radial glia progenitor cells (RGPCs) spawn various cell types including intermediate progenitor cells, neurons, astrocytes, oligodendrocytes, and ependymal cells[1–4]. Following asymmetric cell division at the ventricular surface, the self-renewed RGPCs remain anchored to their neighbors via cadherin-based adherens junctions[5,6]. This adhesion complex not only serves as a structural stabilizer, but it is also involved in important signaling mechanisms regulating cell cycle, proliferation, and differentiation, indicating that adherens junction assembly and disassembly are tightly coupled to cell fate decisions[7–9]. Accordingly, the non-RGPC-fated daughter cells need to break down their adherens junctions to delaminate from the ventricular wall and to migrate to their functional destinations along the radial scaffold of RGPCs[6,9,10].

Appropriately timed delamination is critically important for the mature daughter neuroblasts to follow their normal migratory route[6]. In contrast, pathological detachment and abnormal dispersion of RGPCs, which retain their proliferative capacity at ectopic locations, represent a serious risk for brain malformations, such as focal cortical dysplasia or periventricular heterotopia, potentially predisposing to various forms of intellectual disabilities and neurological deficits[11–14]. Enormous number of cell division events and subsequent delamination steps are required for the generation of hundreds of billions of neurons and other cell types in the brain. Considering the remarkably low prevalence of congenital brain malformations, it is conceivable to hypothesize that a specific mechanism exists that protects against the consequences of sporadic delamination errors. Yet, how pathologically detached RGPCs are eliminated from the brain parenchyma and the related fundamental question of how fate-committed daughter cells become resistant to this defense mechanism both remain elusive.

Along a similar line of reasoning, it is plausible to predict that such a protective mechanism should also be activated when pervasive injuries or teratogenic insults damage the adherens junctions in the prenatal brain. For example, germinal matrix hemorrhage, the most common neurological disease of preterm infants as well as microcephaly-associated environmental teratogens, such as the Zika virus and fetal alcohol exposure are all known to impair cell–cell adhesion in the germinative ventricular zone (VZ) leading to delamination and subsequent depletion of RGPC pools[15–18]. However, the molecular link between adherens junction damage and cell death induced by injury or by teratogenic insults remains unknown.

In the present study, we tested the hypothesis that a yet unidentified safeguarding mechanism determines distinct cell fate after normal delamination of daughter cells or pathological detachment of progenitor cells. We demonstrate that adherens junction impairment induces aberrant RGPC delamination, ectopic accumulation, and caspase-mediated apoptosis in the subventricular zone (SVZ) and VZs, whereas caspase inhibition not only prevents the evoked cell death, but also rescues radial migration into the cortical plate. Moreover, abhydrolase domain containing 4 (ABHD4), a serine hydrolase with a yet undefined neurobiological function is identified as a necessary and sufficient mediator for the elimination of pathologically detached cells. Accordingly, healthy neuroblasts delaminating according to their normal developmental program switch off ABHD4 expression in parallel with their neurogenic commitment. ABHD4 is not required for the canonical form of developmentally controlled programmed cell death in the embryonic neocortex indicating that its function is specifically associated with pathological insults. In agreement with this possibility, our findings elucidate that ABHD4 is also essential for prenatal alcohol exposure-induced apoptosis providing insights into the mechanisms underlying cell death associated with fetal alcohol syndrome.

## Results

**Pathological RGPC detachment triggers developmental anoikis.** N-cadherin (encoded by the *Cdh2* gene) is the major molecular component of the adherens junction belt along the ventricular wall in the developing mammalian brain[5]. To interfere with cadherin-based cell-cell adhesions, we carried out in utero electroporation of a dominant-negative version of N-cadherin (*ΔnCdh2-GFP*) into the lateral ventricles at embryonic day 14.5 (E14.5). We exploited this conditional and sparse adherens junction injury protocol instead of the global loss-of-function approaches to restrict the effects in time and space to a selected population of RGPCs and to avoid potential compensatory mechanisms that are more likely to come into play upon systems-level perturbations. Accordingly, in utero electroporation (IUE or EP) of *ΔnCdh2-GFP* caused a destruction of adherens junctions limited to the electroporated area (Fig. 1a–d; for comparison of electroporated and non-electroporated area see Supplementary Fig. S1a–f). Confocal and STORM super-resolution microscopy revealed a striking specificity of this experimental manipulation as basal processes of electroporated RGPCs still reached the basal surface in *ΔnCdh2*-electroporated cortices (Fig. 1e–j). Moreover, the morphology of the basal endfeet at the pial surface (Fig. 1g, j, k) and the nanoscale architecture of radial scaffold processes of transfected RGPCs both remained unaltered 24 h later (Supplementary Fig. S1g–n). As a result of the adherens junction disruption, the detached RGPCs delaminated from the ventricular surface and accumulated in the SVZ. Most of these pathologically detached progenitor cells retained PAX6 transcription factor expression, a marker of RGPCs[19] even outside of the ventricular germinative niche (Fig. 1l–p). Immunostaining for the mitotic marker phospho-histone H3 (PHH3) revealed that the dispersed RGPCs are still undergoing proliferation (Supplementary Fig. S2a–d). In agreement with prior data that some RGPCs start differentiating as a result of adherens junction loss induced by *ΔnCdh2*-electroporation[7], we also noticed an increase in the ratio of presumptive basal radial glia cells (bRG, based on the presence of the basal process in SVZ-located mitotic cells) and intermediate progenitor cells (IP, TBR2-positive cells with no apical or basal extensions in the SVZ) at the expense of RGPCs/apical radial glia precursors (mitotic cells in the VZ with both apical and basal processes; Supplementary Fig. S2e–i, j) together with a switch of fate based on the altered PAX6- and TBR2-immunostaining patterns (Supplementary Fig. S2k–q).

In order to investigate the fate of these pathologically detached RGPCs, we next tested whether cell death is also induced by adherens junction disruption. Notably, *ΔnCdh2-GFP*-electroporation caused a strong increase in the density of dead cells visualized by TUNEL-labeling when compared to *GFP*-electroporation in control animals (Fig. 2a, b, c, d, k, l, m, n, o, t). Although dying cells lose their protein content and consequently their immunogenicity, the majority of TUNEL-positive cells (73%; 232/317 cells) still had a relatively normal cell-like morphology and expressed PAX6. As a minimal estimation, ~30% of all TUNEL-positive signals overlapped with PAX6-immunostaining, indicating that many cells already arrived to a later stage of apoptosis (Fig. 2e–k). In addition, administration of the pan-caspase inhibitor Z-VAD-FMK fully prevented the increase in cell death evoked by adherens junction disruption (Fig. 2l–t). In contrast, basal cell death levels remained unaffected demonstrating that pathological detachment specifically triggers a caspase-dependent apoptotic process (Fig. 2l, m, p, q, t). Moreover, the ectopic accumulation of RGPCs in the SVZ was also rescued by Z-VAD-FMK treatment and most of the surviving *ΔnCdh2-GFP*-electroporated cells could migrate to the cortical plate (Supplementary Fig. S3). These observations reinforce the idea that a specific cell death mechanism exists to

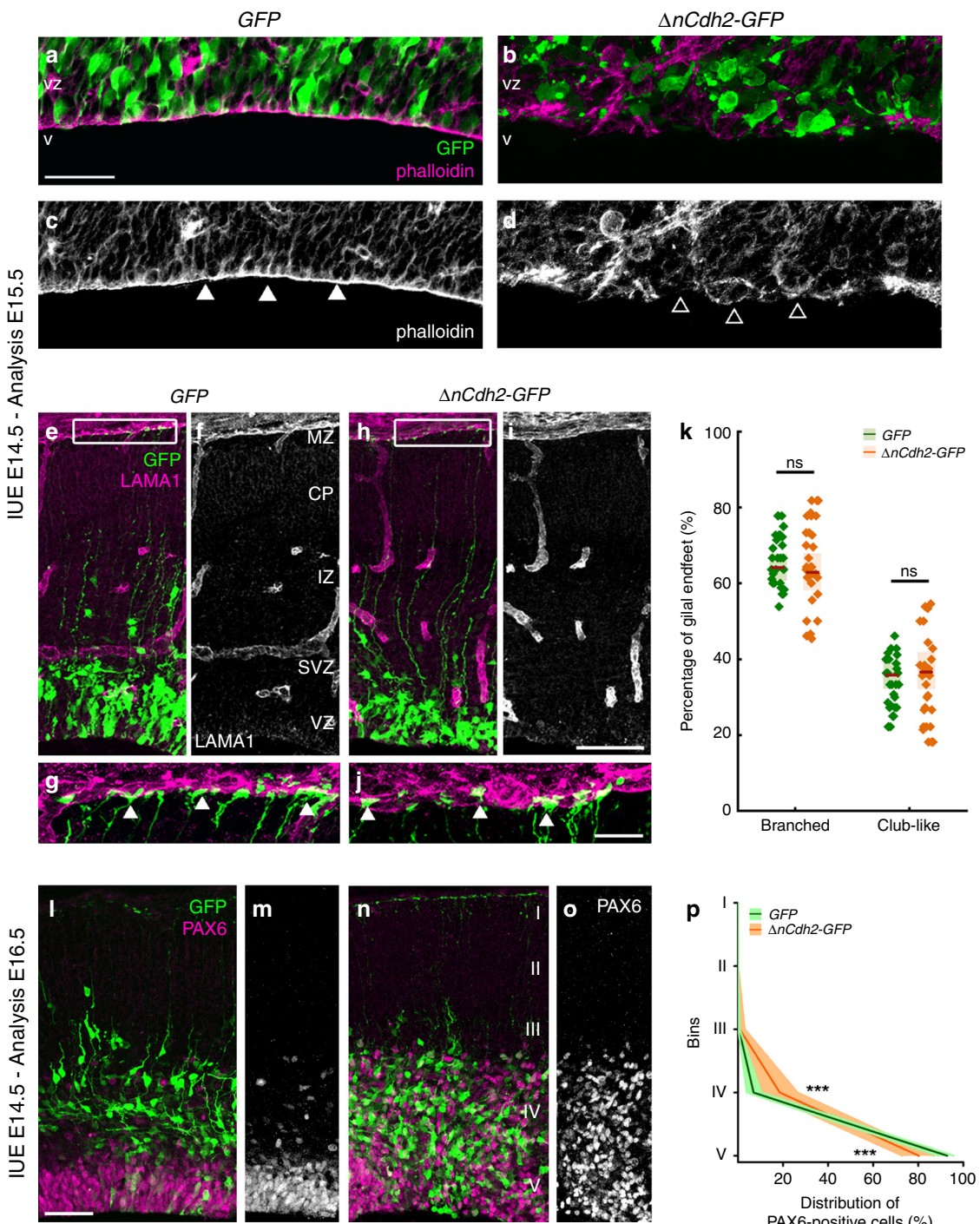

**Fig. 1 Adherens junction disruption induces ectopic accumulation of PAX6-positive cells.** High-power fluorescent micrographs of the ventricular zone (VZ) show the adherens junction belt (white arrowheads) around the ventricle (V). *ΔnCdh2-GFP*- (**b, d**), but not control GFP- in utero electroporation (IUE) (**a, c**), demolishes adherens junctions (open arrowheads). **e–j** Laminin (LAMA1)-immunostaining of the developing cerebral cortex from *GFP*- and *ΔnCdh2-GFP*-electroporated embryos. High-power images (**g, j**) demonstrate that the attachment of the basal endfeet of RGPCs to the pial surface is not affected directly by the disruption of cadherin-based adherens junctions at the ventricular surface. **k** Selective damage of the apical adherens junction does not affect the two main morphological types of radial glia endfeet at the basal surface (two-sided Student's unpaired *t* test, *P* = 0.951; *n* = 43 sections from *n* = 3 animals per GFP treatment, *n* = 37 sections from *n* = 3 animals per *ΔnCdh2-GFP*-treatment). Graphs show box-and-whisker plots (including minima, maxima and median values, lower and upper quartiles) with single values. **l–o** Breakdown of adherens junctions prompts delamination (**g–h**) and ectopic accumulation of PAX6-positive cells in the subventricular zone. **p** Distribution of PAX6-positive cells in five equal bins (Roman numerals) (two-sided Mann–Whitney *U* test for all comparisons; 4th bin ***P* = 0.0004 and 5th bin ***P* = 0.0003; *n* = 13 sections from *n* = 3 animals per GFP treatment; *n* = 11 sections from *n* = 3 animals per *ΔnCdh2-GFP*-treatment). Data are shown as median (line) and interquartile range (transparent band). Scale bars: **a–d, l–o**: 50 μm, **e–j**: 25 μm. Source data are provided as a Source Data file.

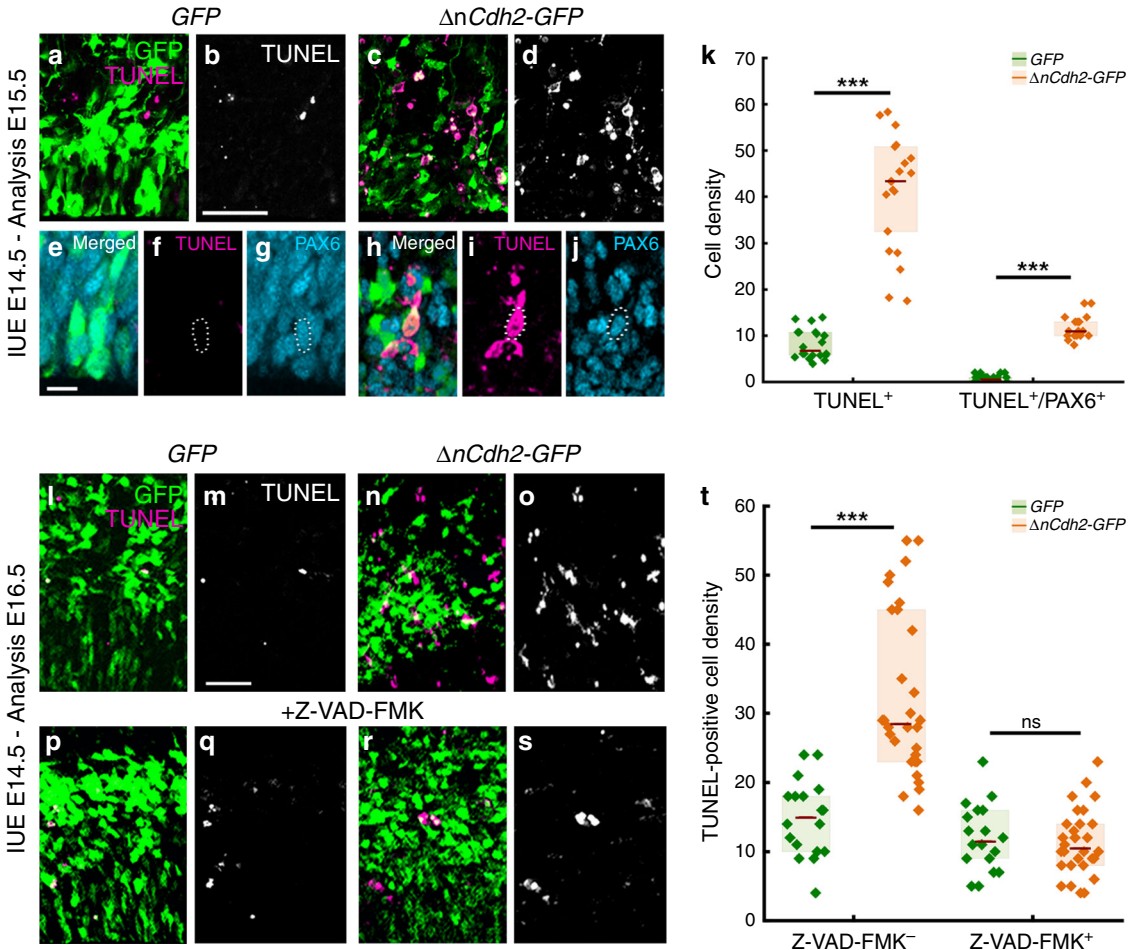

**Fig. 2 Adherens junction disruption induces apoptosis in the prenatal neocortex. a–d** Confocal images demonstrate that *ΔnCdh2-GFP-* (**c**, **d**) triggers increased cell death compared to control *GFP*-electroporation (**a**, **b**). **e–g** GFP-electroporated PAX6-expressing cells occasionally show TUNEL-labeling under control conditions. **h–j** Increased TUNEL-labeling density is observed in *ΔnCdh2*-electroporated samples. Scattered line encircles a still morphologically intact *ΔnCdh2*-electroporated, TUNEL-positive cell that has retained PAX6-immunostaining. **k** Density of TUNEL-positive and TUNEL/ PAX6 double-positive cells after electroporation at E15.5 (both cases: two-sided Mann–Whitney *U* test, *P* < 0.0001; *n* = 22 samples from *n* = 3 animals per *GFP* electroporation, *n* = 21 samples from *n* = 3 animals per *ΔnCdh2-GFP* electroporation). **l–o** Two days after the elimination of adherens junctions show elevated cell death in the electroporated area (**n**, **o**). **p–s** The pan-caspase inhibitor Z-VAD-FMK prevents cell death induced by *ΔnCdh2-GFP*-electroporation. **t** Density of TUNEL-positive dead cells in the electroporated area (Kruskal–Wallis test with post hoc Dunn's Test; ***P* < 0.0001; ns = not significant, *P* ≈ 1; *n* = 18–18 sections from *n* = 3–3 animals per *GFP* and *GFP* + Z-VAD-FMK treatment; *n* = 30–30 sections from *n* = 4–4 animals per *ΔnCdh2-GFP* and *ΔnCdh2-GFP* + Z-VAD-FMK treatment). Graphs show box-and-whisker plots (including minima, maxima, and median values, lower and upper quartiles) with single values. Scale bars: **a–d**, **l–s**: 50 μm, **e–j**: 10 μm. Source data are provided as a Source Data file.

eradicate pathologically detached progenitor cells in the developing neocortex, and this mechanism must be switched off in normally delaminating fate-comitted daughter cells to permit their migration to their functional destinations. We termed this process developmental anoikis, because it conceptually resembles the specific form of apoptosis of epithelial cells induced by the loss of cell anchorage[20,21]. Anoikis has primarily been implicated as a protective mechanism in tumor biology, and pathologically detached tumor cells need to develop resistance to anoikis to become able to invade distant organs during metastasis formation[22].

**Selective ABHD4 expression in RGPCs in the prenatal brain.** The previous experiments led us to hypothesize that specific molecular players have evolved to mediate delamination error-induced cell death. These signaling components must be present in anchorage-dependent RGPCs to protect them from pathological insults, but their expression should be downregulated in migrating healthy neuroblasts. Therefore, we searched public

single-cell RNA-Seq databases, and identified abhydrolase domain containing 4 (ABHD4), an endocannabinoid-metabolizing enzyme[23] with a yet unknown in vivo function, as a potential molecular candidate. Notably, *Abhd4* mRNA levels were below detection threshold in more committed neuronal progenitor cell populations and in adult cortical neuronal types[24,25], whereas *Abhd4* was found to be highly expressed in putative RGPC pools in both mouse and human embryonic cortical samples and cerebral organoids[26,27]. The pattern of expression was very similar to the RGPC marker *Pax6*, but was complementary to that of *Tbr1*, a marker of differentiated neurons (Supplementary Fig. S4). Moreover, a target-agnostic in vitro shRNA library screen in immortalized prostate epithelial cells suggested that downregulation of ABHD4 may potentially contribute to the resistance to anoikis[28].

To experimentally determine whether the expression pattern of ABHD4 matches with its potential functional role in the developing neocortex, we first performed in situ hybridization on wild-type and littermate *Abhd4*-knockout embryonic brains.

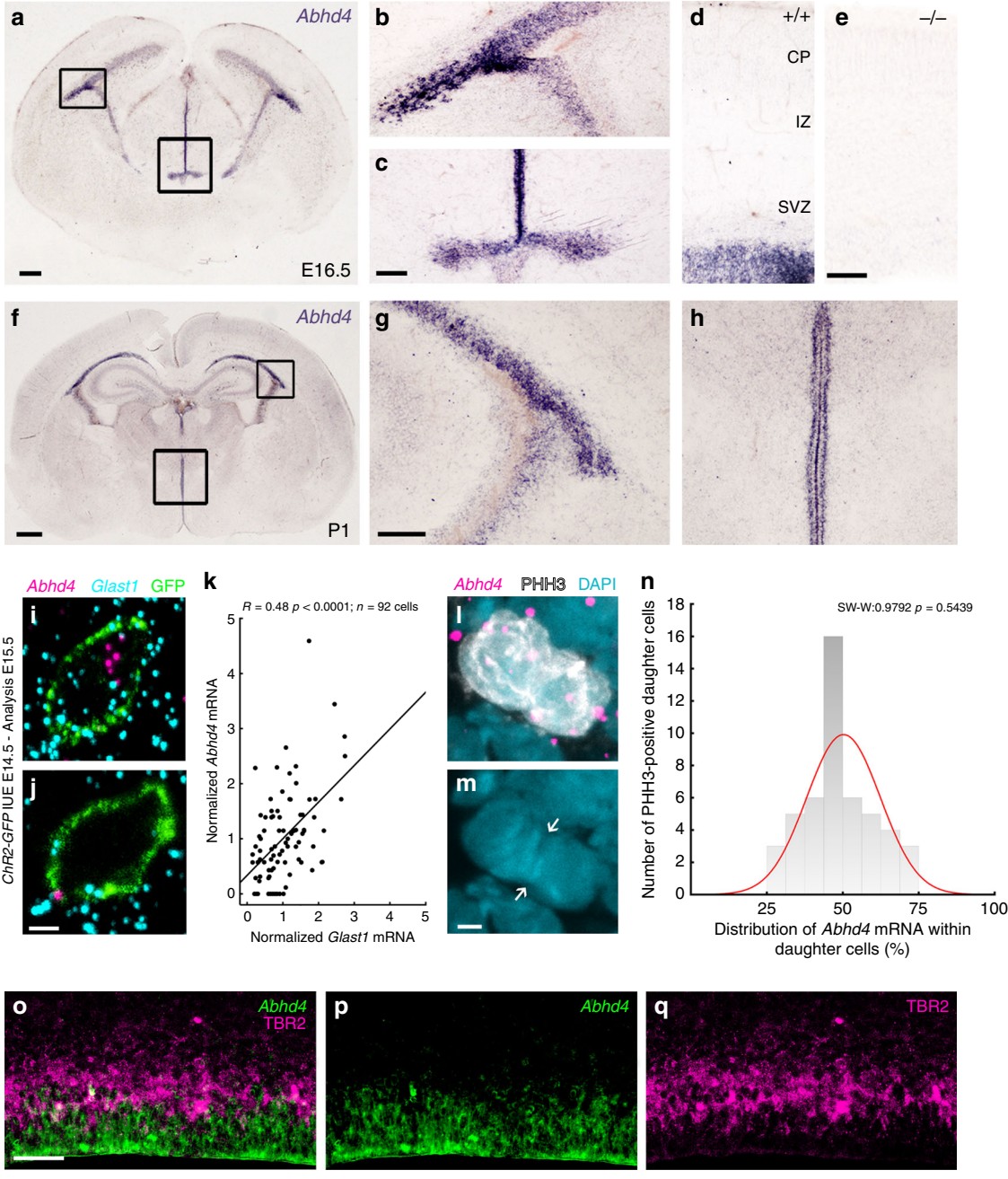

**Fig. 3 *Abhd4* mRNA is expressed by radial glia progenitor cells. a-h** *Abhd4* mRNA is present exclusively in the ventricular zone along with the lateral (**b**, **g**) and third ventricles (**c**, **h**) at both E16.5 (**a-d**) and P1 (**f-h**) wild-type (+/+) mice. The specificity of the *Abhd4* riboprobe is validated in *Abhd4*-knockout (−/−) animals (**e**). CP, cortical plate; IZ, intermediate zone; SVZ, subventricular zone; VZ, ventricular zone. High-power confocal imaging outlines the plasma membrane of *ChR2-GFP*-electroporated cells and delimits multi-color RNAscope analysis into single cells within the heterogeneous and densely packed cell layer of the ventricular zone. *Abhd4* mRNA typically colocalizes with the radial glia progenitor cell marker *Slc1a3* mRNA (encoding GLAST1 protein) (**i**), whereas other cells are often devoid of both markers (**j**). **k** Correlation analysis of *Abhd4* mRNA levels with *Glast1* mRNA levels in single cells (Spearman's rank correlation, *Abhd4/Glast1*: R = 0.48, P < 0.0001; n = 92 cells from n = 4 mice). The scatter plot shows data from individual cells normalized to the median value of the respective mRNA levels. **l**, **m** *Abhd4* mRNA distribution in attached daughter cells marked by PHH3-immunostaining. Arrows point to the mitotic cleavage furrow between the dividing cells. **n** Quantification of *Abhd4* mRNA allocation within PHH3-positive daughter cells (Shapiro-Wilk normality test; W = 0.9792; P = 0.5439; n = 24 sections from n = 3 animals). **o-q** Representative images for *Abhd4* in situ hybridization combined with TBR2-immunostaining. *Abhd4* mRNA shows complementary distribution to TBR2 protein-containing intermediate progenitor cells. Scale bars: **a**: 100 μm, **b-e**, **g-h**, **o-q**: 50 μm, **f**: 500 μm, **i**, **j**, **l**, **m**: 2 μm. Source data are provided as a Source Data file.

These experiments revealed that *Abhd4* mRNA expression was remarkably abundant in the germinative niches of the telencephalic and third brain ventricles, whereas it was absent in other regions and in control *Abhd4*-knockout embryonic brains (Fig. 3a–e). This spatially restricted expression pattern was consistent throughout prenatal development (Supplementary Fig. S5a–f), but *Abhd4* expression markedly decreased postnatally in parallel with the reduced number of proliferating progenitors in the subventricular and subgranular zones (Fig. 3f–h; Supplementary Fig. S5g–i), reaching undetectable levels in adults.

Immunoblotting with a specific antibody raised against a conserved disordered motif of the ABHD4 protein further confirmed the presence of this serine hydrolase enzyme in the developing neocortex of wild-type, but not of *Abhd4*-knockout control embryos (Supplementary Fig. 5Sj, k).

Although RGPCs represent the majority of cells in the germinative niches, it is important to note that fate-committed daughter cells that are undergoing delamination still populate the VZ, where the high cellular abundance renders cell-specific quantitative mRNA analysis very difficult. In order to unequivocally identify the cell population expressing *Abhd4*, we developed an approach for single cell-restricted, high-resolution visualization of the plasma membrane via in utero electroporation combined with multi-color in situ hybridization (see "Online "Methods"). We found that *Abhd4* mRNA levels were positively correlated with *Glast1* expression (a marker of RGPCs[29]; Fig. 3i, j). To test the possibility that *Abhd4* mRNA is preferentially segregated either into self-renewing RGPCs or daughter cells during cell division, we also measured *Abhd4* expression by quantifying RNAscope in situ hybridization signal within mitotic cells visualized by PHH3-immunostaining. The distribution analysis yielded a single-peak Gaussian curve with a peak value around 50% indicating the uniform spatial distribution of *Abhd4* mRNA within mitotic cell pairs (Fig. 3l–n). Subsequently, a combination of *Abhd4* in situ hybridization with TBR2-immunostaining convincingly demonstrated the lack of overlap of *Abhd4* mRNA expression with TBR2 protein, a marker of delaminated IP cells (Fig. 3o–q). These findings together indicate that the temporal expression of *Abhd4* in RGPCs is tightly controlled and its downregulation in fate-committed daughter cells starts rapidly after their delamination.

**ABHD4 is sufficient to trigger apoptosis**. The previous experiments demonstrated strict spatial and temporal restriction of *Abhd4* expression in RGPCs of the developing brain. RGPCs serve two major functions in the embryonic cerebral cortex: as progenitor cells[1–3] and by providing a scaffold for postmitotic neuroblast migration[10]. By using *Abhd4*-knockout mice, we first tested the possibility that ABHD4 plays a role in RGPC proliferation. However, neither single-pulse bromodeoxyuridine-labeling of proliferating precursors (Fig. 4a–c), nor direct visualization of mitotic cells by PHH3-immunostaining (Supplementary Fig. S6a–c) revealed any quantitative differences between wild-type and *Abhd4*-knockout mice. This corresponds well with the lack of any changes in the number of PAX6-positive RGPCs (Fig. 4d–f) or in the structure of the adherens junction belt in the ventricular wall (Fig. 4g, h), together indicating an intact and functional neurogenic niche in *Abhd4*-knockout mice. Delamination and migration did not require ABHD4 either, because there was no change in the number and laminar distribution of TBR2-immunopositive intermediate progenitor cells[30] in the SVZ, and the density of TBR1-immunopositive postmitotic neurons[31] in the cortical plate of the *Abhd4*−/− embryonic cerebral cortex (Supplementary Fig. S6d–i). Moreover, the laminar distribution of pyramidal cells, GABAergic interneurons and astrocytes was also unaltered in the adult neocortex of knockout animals (Supplementary Fig. S7). Finally, quantitative STORM microscopy revealed that the nanoarchitecture of the radial glia processes also remained intact in the absence of ABHD4 (Fig. 4i–p). Together, these results demonstrate that despite its high expression in RGPCs, ABHD4 is not necessary for the two major developmental functions of these progenitor cells.

These unexpected findings, together with the spatially and temporally restricted expression of *Abhd4* in RGPCs, raise the possibility that this serine hydrolase has evolved to fulfill a hitherto undefined physiological function that is important in the cell biology of neurogenesis. Interestingly, phylogenetic analysis revealed that ABHD4 protein orthologues exhibit substantial homology that is especially high around the catalytic serine residue (S159 in mouse[23]) and the consensus nucleophile elbow sequence GXSXG, suggesting that the enzymatic function of ABHD4 is evolutionarily conserved (Supplementary Fig. S8).

Because *Abhd4* expression is rapidly downregulated in delaminated daughter cells right after their fate-commitment (Fig. 3o–q), we reasoned that counteracting this process by ectopic *Abhd4* expression in these cells could shed light on the conserved function of ABHD4 in the developing brain. To test this idea, we in utero electroporated either *GFP* alone, or wild-type *Abhd4-GFP*, or a catalytically inactive form of *Abhd4-GFP* (in which the catalytic serine residue that is required for hydrolase activity was mutated to glycine, S159G mutant) at E14.5. Interestingly, ABHD4, but not the inactive, hydrolase-dead form of ABHD4 caused a striking migration arrest within the SVZ similar to that of found after adherens junction disruption (Fig. 5a–d). Moreover, *Abhd4-GFP*-transfected cells lost the characteristic bipolar morphology of migrating neuroblasts and became shrunken, rounded, and had no leading processes (Fig. 5e–h). These morphological changes are hallmarks of imminent cell death. Indeed, we observed that ABHD4 alone, but not its inactive form could trigger a robust (295%) increase in caspase-dependent apoptosis (Fig. 5i–o) when compared to *GFP*-electroporated controls, indicating that the enzymatic activity of ABHD4 is sufficient to elicit apoptosis in the delaminated cells.

A shift in the balance of proapoptotic and pro-survival molecular mechanisms is critically important in anoikis-related cell fate decisions during transition of cells to a metastatic phenotype[32]. We reasoned that some daughter cells whose transfection coincided with their delamination should already express the essential pro-survival molecular toolset to escape from anoikis and may be capable to balance the impact of exogenous *Abhd4* over-expression by its direct downregulation. To test this possibility, we next co-electroporated tdTomato either with a fusion protein construct (*Abhd4-GFP-F*) or with the bicistronic construct (*Abhd4-GFP*). Notably, we found that initially GFP expression in case of both of these constructs almost completely overlapped with the co-electroporated tdTomato marker 24 h after co-electroporation (Supplementary Fig. S9a–f). In striking contrast, only tdTomato-expressing, but GFP-negative cells were observed with the fusion construct in layer 2/3 in long-term survival experiments, whereas tdTomato extensively colocalized with GFP in case of the control bicistronic construct from which translation of ABHD4 and GFP occurs separately (Supplementary Fig. S9g–k). This observation suggests that only those postmitotic neurons could undergo delayed migration by postnatal day 3 that could degrade their ABHD4-GFP fusion protein and indicate that ABHD4 activity is non-permissive for radial migration into the cortical plate.

To determine how powerfully ABHD4 drives cell death even without a native tissue context (e.g., in the absence of additional proapoptotic signals), we next examined Human Embryonic Kidney-293 cells, which are relatively resistant to proapoptotic stimuli due to their deregulated pRB/p53 pathway and also lack endogenous *Abhd4* expression[33]. Correlated confocal and STORM super-resolution imaging revealed that ABHD4, but not its inactive form caused a substantial loss of TOM20-positive mitochondria (Supplementary Fig. S10a–f, g) and a concomitant rise in extra-mitochondrial cytochrome c levels, signs of early stage apoptosis (Supplementary Fig. S10e, f, h–j). Moreover, ABHD4 also elicited a large increase in the number of cleaved caspase 3-immunopositive cells exhibiting nuclear fragmentation and condensation, features of late stage apoptosis (Supplementary

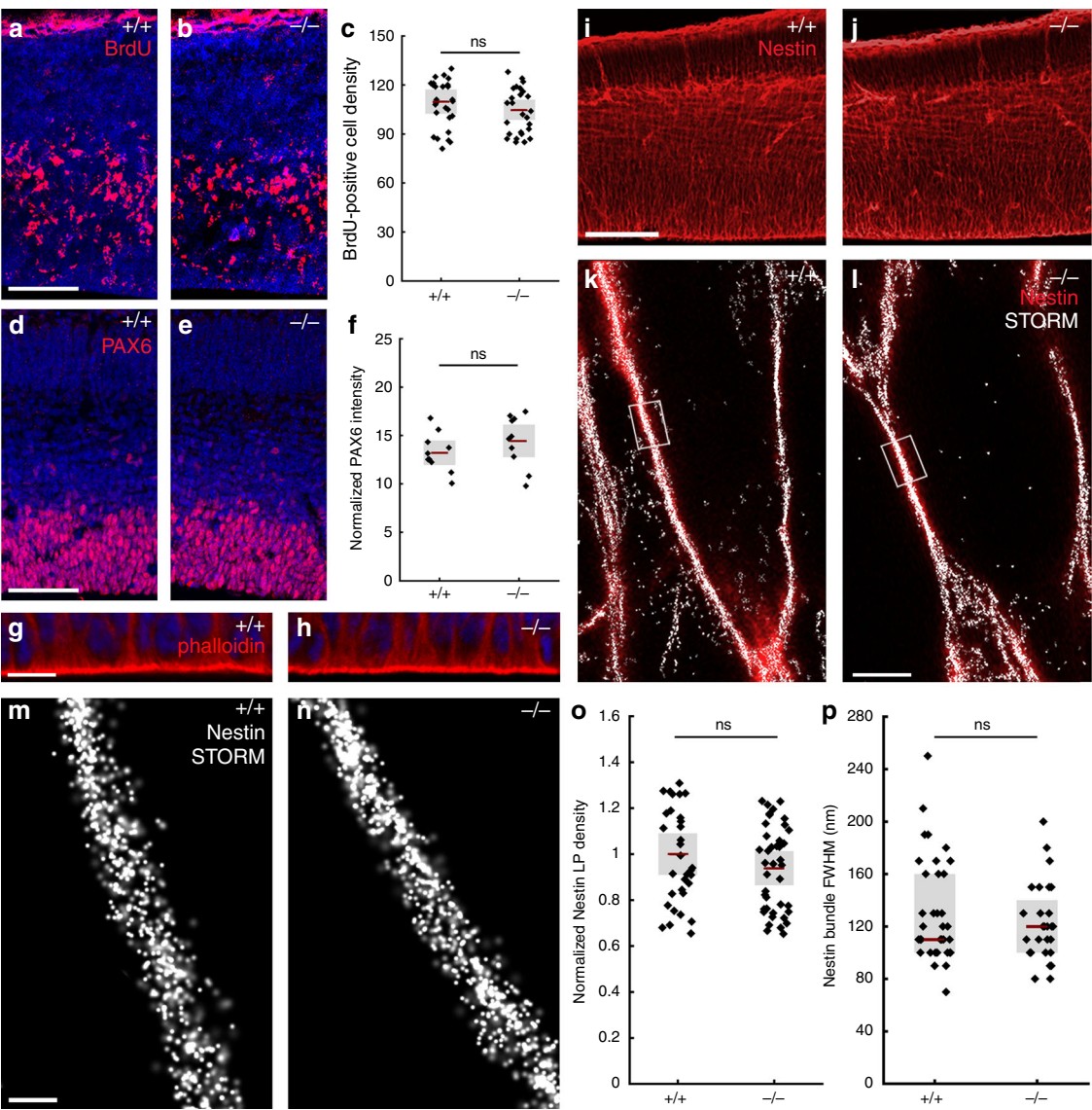

**Fig. 4 ABHD4 is not required for classical radial glia progenitor cell functions. a, b** Coronal sections of the embryonic cortex at E.14.5 show replicating cells that incorporated bromodeoxyuridine (BrdU) during S phase of the cell cycle are shown. **c** Quantification of BrdU-positive cell density (two-sided Student's unpaired t test, P = 0.323; n = 38 sections from n = 6 animals per wild-type (+/+) mice, n = 29 sections from n = 4 animals per *Abhd4*-knockout (−/−) mice). **d, e** Confocal images and **f** quantification of PAX6-positive radial glia progenitor cells in the ventricular zone. (two-sided Student's unpaired t test, P = 0.2598; n = 10 sections from n = 4 animals per wild-type (+/+) mice, n = 10 sections from n = 4 animals per *Abhd4*-knockout (−/−) mice). **g, h** Identical pattern of phalloidin-labeling of the adherens junction belt in the ventricular zone (VZ) of wild-type (+/+) and *Abhd4*-knockout (−/−) mice at E15.5. **i, j** Low-power confocal images show similar organization of nestin-positive radial processes in both genotypes at E15.5. **k, l** Correlated confocal and STORM super-resolution microscopy images from both genotypes. **m, n** STORM super-resolution images reveal that the nanoarchitecture of nestin intermediate filament bundles at the core of the radial glia processes remains intact in the absence of ABHD4. **o** Density of localization points (LPs) representing nestin (two-sided Student's unpaired t test, P = 0.297; n = 39 segments from n = 3 animals per wild-type (+/+) mice, n = 57 segments from n = 3 animals per *Abhd4*-knockout (−/−) mice). **p** Quantification of full-width-at-half-maximum of nestin filament bundles (two-sided Mann–Whitney U test, P = 0.684; n = 28 segments from n = 3 animals per wild-type (+/+) mice, n = 37 segments from n = 3 animals per *Abhd4*-knockout (−/−) mice). Graphs show box-and-whisker plots (including minima, maxima and mean values with 2 × standard error, except **p** where median and lower, upper quartiles are presented) with single values. Scale bars: **a**–**e**, **i**–**j**: 50 μm, **g**–**h**: 20 μm, **m**, **n**: 200 μm. Source data are provided as a Source Data file.

Fig. S10k–q). In contrast, the adjacent untransfected cells remained unaffected (Supplementary Fig. S10r). These results indicate that ABHD4 is capable of triggering an apoptotic program cell-autonomously even in cell types with a higher apoptotic threshold.

**ABHD4 is necessary for developmental anoikis.** It is well-established that cell type-specific consecutive waves of

developmentally controlled programmed cell death delete a substantial proportion of neurons during cortical development[34–36]. Considering the proapoptotic activity of ABHD4, we therefore investigated the basal level of cell death in the developing neocortex. Notably, there was no difference in the density of dead cells in the subventricular and VZs of wild-type and *Abhd4*-knockout mice at embryonic day 16.5 (Fig. 6a–e). Developmentally controlled programmed cell death of glutamatergic cells peaks early postnatally[35] but we could detect no differences

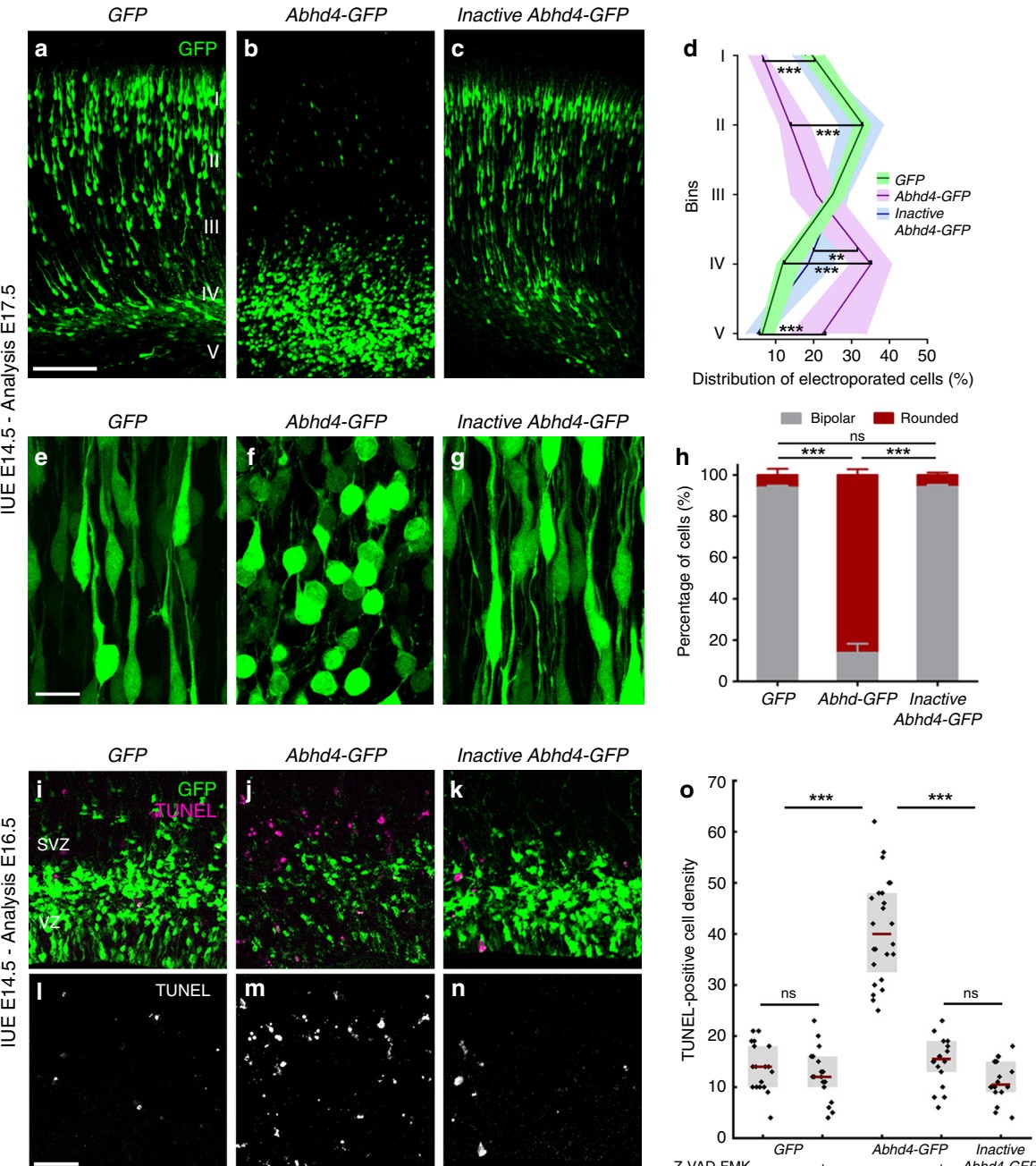

**Fig. 5 ABHD4 is sufficient to trigger migration arrest and apoptosis.** Cells in utero electroporated with *GFP*- (**a**), or with an *Abhd4-GFP*-construct encoding an enzymatically inactive form of ABHD4 (**c**) migrate into the cortical plate, whereas *Abhd4-GFP*-electroporation causes radial migration defect (**b**). **d** Laminar distribution of electroporated cells in five equally-sized bins (Roman numerals) (Kruskal–Wallis test with post hoc Dunn's test, 1st, 2nd, 4th, 5th bins for all comparisons: ***P < 0.0001, except 4th bin (*Inactive Abhd4-GFP* vs *Abhd4-GFP*): **P = 0.006; n = 19 sections from n = 4 animals per *GFP*-electroporation; n = 18 sections from n = 4 animals per *Abhd4-GFP*-electroporation; n = 12 sections from n = 3 animals per Inactive *Abhd4-GFP*-electroporation). Data are shown as median (line) and interquartile range (transparent band in the same color). High-power images show bipolar migrating neurons (**e**, **g**), that become rounded upon *Abhd4-GFP*-electroporation (**f**). **h** Quantification of cell morphology (Kruskal–Wallis test with post hoc Dunn's test, ***P < 0.0001; ns = not significant, P ≈ 1; n = 15–15 sections from n = 3–3 animals per treatment). Graphs show bar plots with median ± interquartile range. **i–n** Representative images show large density of TUNEL-positive cells upon *Abhd4-GFP* electroporation in the subventricular (SVZ) and ventricular (VZ) zone (**j**, **m**), that is not replicated by *GFP*- (**i**, **l**), or by inactive *Abhd4-GFP*-electroporation (**k**, **n**). **o** Quantification of TUNEL-positive cell density (Kruskal–Wallis test with post hoc Dunn's test, ***P < 0.0001; ns = not significant, P ≈ 1 except per *Abhd4-GFP*-electroporation with Z-VAD-FMK treatment vs *Inactive Abhd4-GFP* P = 0.607; n = 18–18 sections from n = 3–3 animals per *GFP*-electroporation with and without Z-VAD-FMK treatment and per Inactive *Abhd4-GFP*-electroporation; n = 24 sections from n = 4 animals per *Abhd4-GFP*-electroporation; n = 18 sections from n = 4 animals per *Abhd4-GFP*-electroporation with Z-VAD-FMK treatment). Graphs show box-and-whisker plots (including minima, maxima, and median values, lower and upper quartiles) with single values. Scale bars: **a–c**: 100 μm, **e–g**: 20 μm, **i n**: 50 μm. Source data are provided as a Source Data file.

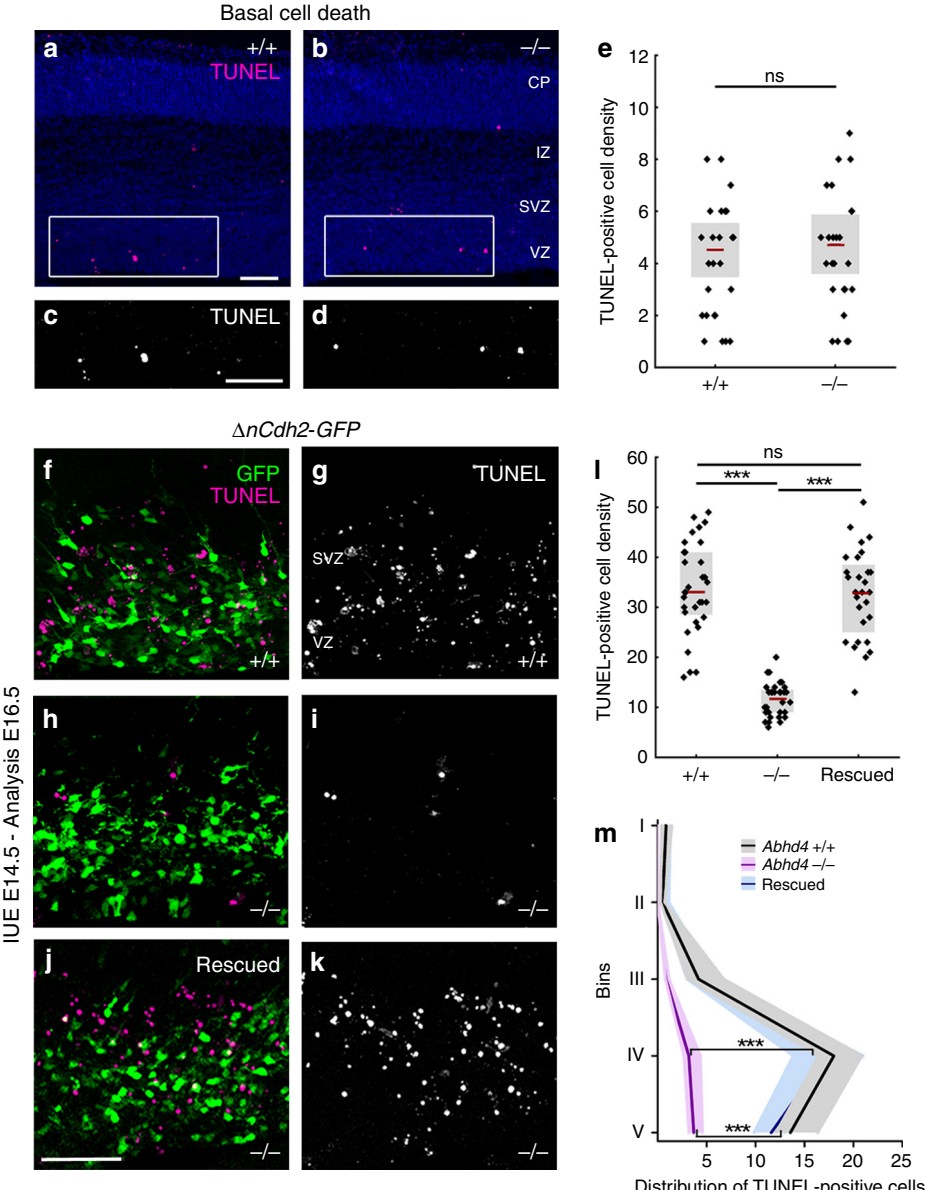

**Fig. 6 ABHD4 mediates apoptosis induced by adherens junction disruption. a–d** Basal level of cell death is similar in coronal sections of the embryonic neocortex of wild-type (+/+) and *Abhd4*-knockout (-/-) mice at E16.5. CP, cortical plate; IZ, intermediate zone; SVZ, subventricular zone; VZ, ventricular zone. **e** Quantification of TUNEL-positive cell density in the VZ/SVZ (two-sided Student's unpaired *t* test, *P* = 0.6964; *n* = 30 sample from *n* = 5 animals from both genotypes). **f, g, i, j,** *ΔnCdh2-GFP in utero* electroporation (IUE) triggers cell death in wild-type (**f, i**), but not in *Abhd4*-knockout mice (**g–j**) in the subventricular (SVZ) and ventricular (VZ) zones. **h, k** Re-expression of *Abhd4-GFP* in *Abhd4*-knockout mice is sufficient to rescue impaired *ΔnCdh2-GFP*-induced cell death. **l** Quantification of TUNEL-positive cell density (Kruskal–Wallis test with post hoc Dunn's test, ***P < 0.0001; ns = not significant, P ≈ 1; *n* = 32–32 sections from *n* = 4–4 animals per genotypes, *n* = 28 sections from *n* = 4 animals per *Abhd4*-re-expression treatment). Graphs show box-and-whisker plots (including minima, maxima, and mean values with 2 × standard error, except **l** where median and lower, upper quartiles are presented) with single values. **m** Distribution of TUNEL-positive cells in five equal bins (Kruskal–Wallis test with post hoc Dunn's test, 4th, 5th bins: ***P < 0.0001; *Abhd* +/+ vs. Rescued *P* = 1). Data are shown as median (line) and interquartile range (transparent band in the same color). Scale bars: **a–d**: 100 μm, **f–k**: 50 μm. Source data are provided as a Source Data file.

in cell death levels between the genotypes at postnatal day 3 either (Supplementary Fig. S11).

Importantly, physiologically controlled forms of programmed cell death or apoptosis induced by pathological insults do not necessarily share the same molecular mechanisms. Because our results have revealed that both adherens junction disruption-induced delamination and ectopic ABHD4 enzymatic activity in normally delaminating cells cause similar caspase-dependent cell death in the embryonic neocortex, we next tested the possibility that ABHD4 serves as a specific molecular link between pathological

detachment and subsequent cell death. To this end, *ΔnCdh2-GFP* was electroporated into the lateral ventricles of wild-type and littermate *Abhd4*-knockout embryos. Sparse adherens junction disruption initiated a marked increase in the density of dead cells that was restricted to the VZ and SVZs (see distribution analysis in Fig. 6m) of wild-type mice and was completely absent in *Abhd4*-knockout animals (Fig. 6f, g, h, i, l). Importantly, this increase in cell death could be fully rescued by *Abhd4* re-expression in *Abhd4*-knockout mice (Fig. 6j, k, l). These findings demonstrate that ABHD4 is not required for developmentally controlled

programmed cell death, but it is essential for developmental anoikis, a unique form of cell death triggered by the damage of cadherin-based adherens junctions in the prenatal brain.

**Maternal alcohol exposure causes ABHD4-dependent cell death.** The most common preventable teratogenic insult in humans is alcohol. Fetal alcohol spectrum disorder has a remarkably high prevalence and it is the foremost non-genetic cause of intellectual disabilities. Prenatal ethanol exposure can cause several brain malformations including cortical dysplasia/heterotopias and microcephaly, all indicating impaired migration and neurogenesis[16]. Importantly, ethanol is known to induce disruption of the adherens junction signaling machinery, to reduce the RGPC pool and to increase cell death, including anoikis[15,16,37,38]. Therefore, we hypothesized that ABHD4 may play a role in cell death associated with prenatal alcohol exposure. We tested this idea by applying two independent ethanol administration protocols to pregnant mice. Both a 3-day-long subchronic ethanol administration regime (Fig. 7) and a single

dose of acute ethanol administration (Supplementary Fig. S12a–i) resulted in a substantial (3–6 fold) increase in cell death and led to the scattered distribution of dead cells restricted to the VZs and SVZs in wild-type embryos. We did not detect elevated cell death in the cortical plate suggesting that mostly the proliferating progenitor pool was affected in the embryonic brain (Fig. 7i). In striking contrast, maternal ethanol administration failed to induce a similar increase in cell death in *Abhd4*-knockout littermate embryos (Fig. 7d, h, i; Supplementary Fig. S12d, h, i). Finally, we did not observe any increase in *Abhd4* expression levels after ethanol-exposure (Supplementary Fig. S12j–l). These results strongly suggest that ABHD4 is also indispensable for ethanol-induced cell death in the embryonic brain.

## Discussion

In the present study, we provide evidence for the existence of developmental anoikis, a distinctive mechanism for cell death in the prenatal brain. Our findings also identify ABHD4, an enzyme that has a vital function in this phenomenon by preventing the survival of misplaced cells that are produced by delamination errors or damaged by alcohol exposure (Fig. 8). Various environmental stressors are known to affect the fetal brain and considering the immense number of cell division events and subsequent delamination steps (>10[11]), the frequency of teratogenic and spontaneous delamination errors is likely to be considerably higher than the observed incidence of congenital brain anomalies (~0.25%, https://eu-rd-platform.jrc.ec.europa.eu/eurocat/eurocat-data/prevalence). Thus, it is conceivable to postulate the pivotal neurological importance of ABHD4-dependent developmental anoikis which eradicates those pathologically detached cells that may represent a risk for future brain malformations, such as dysplasias, heterotopias, or even brain tumors.

It is important to emphasize that RGPCs have a neuroepithelial origin. These proliferating cells maintain most of the epithelial characteristics including the apico-basal polarity via specific molecular anchors to the extracellular matrix and to each other[6].

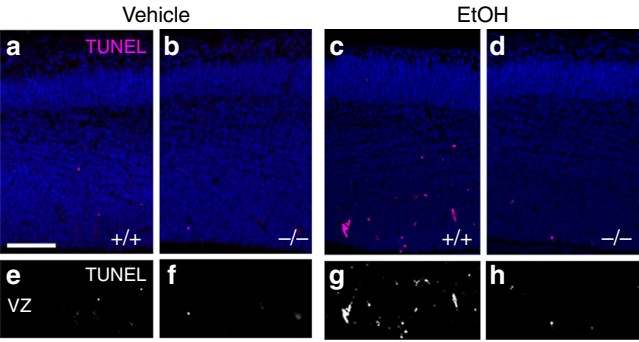

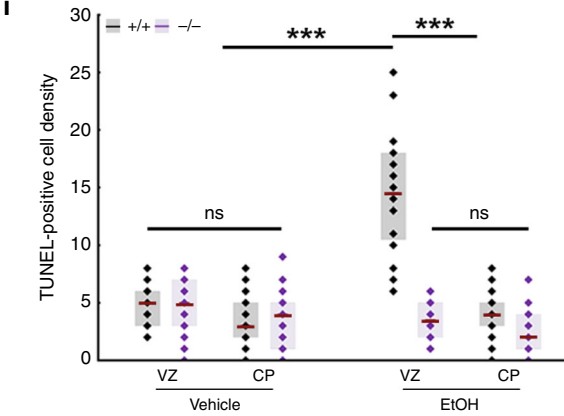

**Fig. 7 ABHD4 is necessary for cell death caused by fetal alcohol exposure.** Compared to vehicle-treated mice (**a**, **b**, **e**, **f**), increased number of TUNEL-positive dead cells is seen in ethanol-treated wild-type (+/+), (**c**, **g**), but not in *Abhd4*-knockout (−/−) mice (**d**, **h**) in the ventricular (VZ) zones of embryos derived from dams undergoing subchronic alcohol treatment, but not in the cortical plate (CP). **i** Quantification of the density of TUNEL-positive cells in the VZ and in the CP after subchronic ethanol-treatment (Kruskal–Wallis test with post hoc Dunn's test, ***$P < 0.0001$; ns = not significant, $P \approx 1$, $n = 31$ sections from $n = 5$ animals per wild-type, vehicle-treated mice, $n = 27$ sections from $n = 4$ animals per *Abhd4*-knockout, vehicle-treated mice, $n = 24$ sections from $n = 4$ animals per wild-type, ethanol-treated mice, $n = 18$ sections from $n = 3$ animals per *Abhd4*-knockout, ethanol-treated mice). Graphs show box-and-whisker plots (including minima, maxima, and median values, lower and upper quartiles) with single values. Scale bars: **a**–**h**: 100 μm. Source data are provided as a Source Data file.

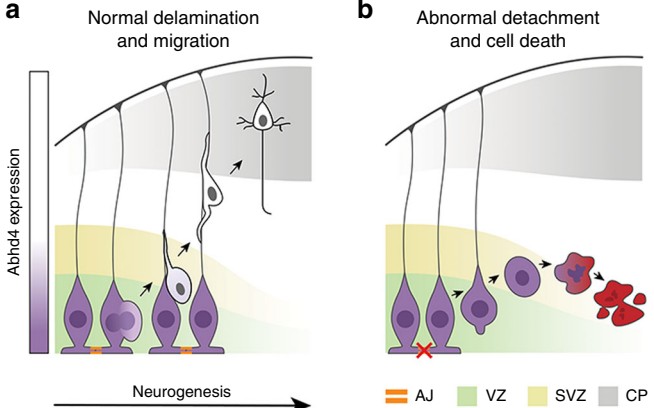

**Fig. 8 Mechanism of ABHD4-dependent developmental anoikis. a, b** Schematic drawing illustrates the distinct ABHD4-dependent fate of normally delaminated and abnormally detached cells in the developing cerebral cortex. After cell division, healthy daughter cells downregulate ABHD4, delaminate from the ventricular wall and migrate into the cortical plate along the radial processes of RGPCs. Random developmental errors and pathological insults such as fetal alcohol exposure often destroy the adherens junctions and trigger pathological detachment. As a central mediator of developmental anoikis, ABHD4 is necessary and sufficient to induce cell death that prevents the accumulation of misplaced progenitor cells at ectopic locations. CP cortical plate, SVZ subventricular zone, VZ ventricular zone.

Right after cell division, the delaminating non-RGPC-fated daughter cell dismantles its cell–cell connections, switches its transcriptional profile and undergoes a characteristic cytoskeletal reorganization to initiate migration. It is increasingly appreciated that several molecular components and the cellular processes that underlie daughter cell delamination are very similar in the so-called epithelial–mesenchymal transition phenomenon[8,39]. Our study further adds to this idea, because the ABHD4-dependent cell death mechanism described here is conceptually analogous to anoikis, a detachment-induced form of apoptosis that has long been considered as the central protective mechanism in epithelial–mesenchymal transition to prevent the proliferation and migration of pathologically detached epithelial cells[20,21,40,41]. Although the term has originally been coined for intrinsic apoptosis induced by the loss of the extracellular matrix-dependent anchorage of epithelial cells[20,21], we propose to call this process developmental anoikis, because of its conceptual similarity and its functional importance during development. Traditionally, anoikis has primarily been implicated as a protective mechanism in tumor biology and obtaining resistance to anoikis is a critical step for cancer cells for tumor invasion and metastasis[21,22,41]. Thus, the results that ABHD4 is a proapoptotic molecule whose expression rapidly disappears in daughter neuroblasts and coincidentally, that ABHD4 downregulation in immortalized prostate epithelial cells prevents cell death[28] together raise the intriguing possibility that a similar mechanism underlies anoikis resistance in healthy migrating neurons and in metastatic cancer cells. From a clinically relevant perspective, another interesting related finding is that ethanol-induced cell death also requires ABHD4 in the embryonic neocortex. Because alcohol turns on the epithelial–mesenchymal transition program that is thought to contribute to the metastatic aggressiveness of epithelial cancers associated with chronic alcohol consumption[42], ABHD4-dependent cell death may play an important role in the attenuation of tumor progression. Notably, data from expression arrays indicate that the *Abhd4* gene is an important downstream target of the tumor suppressor protein p53 in cancer cells at the transcriptional level[43,44]. Thus, it remains to be investigated in future studies whether and how p53 and other related transcriptional regulators contribute to the tightly controlled spatial and temporal expression of *Abhd4* mRNA in the developing brain and under pathological conditions.

At the biochemical level, ABHD4 is involved in N-acyl-ethanolamine and related N-acyl-phospholipid metabolism including the production of the endocannabinoid molecule anandamide[23,45]. Anandamide has various physiological functions including the regulation of synaptic plasticity and apoptosis[46]. Interestingly, NAPE-PLD, another anandamide-synthesizing enzyme is primarily concentrated in presynaptic axon terminals[47] and its enzymatic activity only appears in the postnatal brain[48]. In contrast, our findings demonstrated that ABHD4 is present in radial glia progenitor cells in the prenatal brain, but it disappears postnatally. Together with the apoptosis-triggering function of ABHD4, it is plausible to propose that the multiple enzymatic routes of anandamide synthesis may have evolved to maintain a division of labor of anandamide signaling in time and space. It is likely that in the context of these two very distinct biological phenomena, synaptic plasticity and apoptosis, anandamide signaling requires different regulatory mechanisms, which can be achieved by independently controlling gene expression and enzymatic activity of the different molecular components of the endocannabinoid system.

Beside the potential proapoptotic role of the signaling molecule anandamide produced by ABHD4 activity[46], one must also consider the function of N-acyl-phosphatidylethanolamines (NAPEs), which are the direct substrates of ABHD4[45]. Early biophysical studies reported a stabilizing effect of NAPEs on the integrity of biological membranes and increasing the NAPE content of liposomes inhibits their leakage[49,50]. Thus, ABHD4, as a NAPE lipase may also impair intracellular membrane compartments by hydrolyzing NAPEs. This, combined with anandamide triggering apoptosis via inducing endoplasmic reticulum stress[51] makes it tempting to speculate that the synergistic upstream and downstream legs of ABHD4 activity in combination would represent a very efficient and parsimonious mechanism that renders damaged cells more susceptible to additional proapoptotic processes. In light of the well-described proapoptotic role of phosphatidylserine however, the potential significance of the N-acyl-phosphatidylserines, another substrate class of ABHD4[45], is also worthy of consideration in future studies.

Elucidation of the unique biological function of ABHD4 in the embryonic cerebral cortex will not only help to identify the specific lipid mediators and promote the discovery of additional molecular players cooperating with ABHD4, but will also pave the way to delineate the precise biochemical cascades and biophysical processes of developmental anoikis. This is important, because the present findings indicate that ABHD4-mediated developmental anoikis is a safeguarding mechanism that protects the fetal brain from the effects of delamination errors. In this regard, developmental anoikis can be considered as a phenomenon that contributes to biological robustness that helps to maintain the intactness of brain development against external and internal perturbations[52,53]. The astronomical number of cell divisions combined with the numerous detrimental environmental factors potentially affecting the developing brain puts an enormous stress on progenitor cells and inherently imply a relatively high probability of the occurrence of aberrant cell division and pathological detachment events. Because even occasional delamination errors could cause severe brain malformations due to misplaced proliferation of progenitor cells, protective mechanisms are likely to be very important to counterbalance the risk presented by the immense number of cell divisions and various environmental teratogens. Thus, ABHD4-mediated developmental anoikis may conceptually function similarly to other well-known biological phenomena such as the DNA repair mechanisms, the p53-pathway against genotoxic stress, or the unfolded protein response for protein quality control. In summary, further research on the detailed mechanisms of developmental anoikis will therefore not only gain insights into neurodevelopmental processes underlying congenital brain anomalies and the related neurological and cognitive deficits, but could also extend our knowledge on biological robustness during cortical development.

## Online methods

**Animals**. All experiments were approved by the Hungarian Committee of the Scientific Ethics of Animal Research (license numbers: XIV-1-001/2332-4/2012 and PE/EA/354-5/2018), and were carried out according to the Hungarian Act of Animal Care and Experimentation (1998, XXVIII, Section 243/1998), in accordance with the European Communities Council Directive of 24 November 1986 (86-609-EEC; Section 243/1998). Mice were kept under approved laboratory conditions and all efforts were made to minimize pain and to reduce the number of animals used. Both male and female mice were used throughout the study. C57BL/6 and CD-1 mouse lines were obtained from Charles River Laboratories. Mice bearing a disruption in the *Abhd4* gene were generated from the 129S6/SvEvTac strain and were backcrossed into the C57BL/6 background for more than ten generations prior to present experiments[45].

**DNA constructs and cloning protocols**. Mouse N-cadherin (*Cdh2*) was cloned via RT-PCR using E16.5 brain cDNA as template with long-template PCR mix according to manufacturer's instruction (Roche). The dominant-negative version of N-cadherin (*ΔnCdh2*) was created by OliI–HindIII digestion, followed by Klenow fill-in of HindIII-overhang and religation of the plasmid. The final construct was validated by sequencing and encoded a truncated N-cadherin molecule that contains the signal sequence followed by the transmembrane and intracellular domains but lacks the extracellular domain. The analogous mutant form of N-cadherin has been shown to interfere with cell adhesion in cellular assays derived from clawed frog and chicken[54–56].

Mouse *Abhd4* was also amplified by RT-PCR as described above. The amplicon was cloned into pGEMTEasy plasmid (Promega) and sequenced. The construct encoding the inactive *Abhd4* form was created by site-directed mutagenesis of the catalytic serine residue 159 to glycine via PfuI PCR and DpnI digestion of the template plasmid, followed by transformation and clone identification by sequencing. The *ΔnCdh2*, *Abhd4*, and inactive *Abhd4* constructs were subcloned from pGEMT into the pCAGIG mammalian expression vector. The tdTomato construct was cloned by the removal of the GFP-polyA part of the pCAGIG plasmid and replacement with tdTomato-pA[57]. The pCAGIG plasmid was a gift from Connie Cepko (Addgene #11159)[58].

To generate riboprobes for chromogenic in situ hybridization, a shorter 418bp-long fragment of the *Abhd4* gene and a 406 bp-long fragment of *Slc1a3* gene (encoding the astrocyte marker protein GLAST1*)* were amplified by RT-PCR and cloned into pGEMTEasy vector. The primers used for cloning the cDNAs that served as templates for the riboprobes encoded by the *Slc17a7* and *Gad1* genes[59] are added to the Supplementary Information in Table 1.

**In utero electroporation, Z-VAD-FMK and BrdU injections**. Timed-pregnant female mice were anesthetized with avertin (1.25% v/v, Sigma), and uterine horns were exposed. The bicistronic construct harboring the gene-of-interest and an internal ribosome entry site (IRES)-GFP cassette in the pCAGIG expression vector ($1–2 \mu g \mu l^{-1}$ at 1 μl volume) in endotoxin-free water containing Fast Green (1:10,000, Roth) was electroporated into the lateral ventricle of the embryo via a glass capillary at embryonic day 14.5. Electroporation was performed with tweezer electrodes (5 pulses of 40 V for 50 ms at 950 ms intervals) using an SP-3c electroporator (Supertech). After electroporation, the uterine horns were returned into the abdominal cavity, the wall and skin were sutured and embryos were allowed to continue their normal development. To inhibit caspase activity, the pan-caspase inhibitor Z-VAD-FMK (5 μM, BD Biosciences) was injected in a similar manner as above. To identify proliferating cells, BrdU in 0.9% saline ($200 \text{ mg kg}^{-1}$, Sigma) was intraperitoneally injected into pregnant dams at E14.5. The embryos were collected 2 h later and their brains were fixed with 4% paraformaldehyde (PFA).

**Chromogenic and cell-specific RNAscope fluorescent in situ hybridization**. Embryonic brains (E14.5, E16.5, E18.5) and early postnatal (P1, P3) brains were removed from the skull and immersion-fixed with 4% PFA overnight. Older postnatal (P10) and adult (P60) wild-type and littermate *Abhd4*-knockout mice were transcardially perfused with 4% PFA and their brains were postfixed for 3 h at 4 °C. After fixation, 50-μm-thick free-floating sections were cut with VT-1200S Vibratome (Leica). Chromogenic in situ hybridization using digoxygenin-labeled antisense

riboprobes was performed as explained in a previous study[60]. Chromogenic in situ hybridization combined with fluorescent immunohistochemistry was performed by developing the in situ hybridization signal and then incubating the sections in 10 mM citric acid for one hour at 65 °C. The sections were washed in 0.1% Triton X-100 in PBS before the blocking step and incubated with rabbit antibody to T-box, brain, 2 (TBR2, 1:500, Abcam). The chromogenic signal was detected with a CCD camera, followed by confocal imaging of immunofluorescence signal in the same frame and the two images were overlaid.

The very high cellular abundance in the VZ renders quantitative measurement of mRNA molecules in individual cells very difficult by conventional in situ hybridization approaches. In order to visualize the plasma membrane of a sparse population of cells in the VZ by in utero electroporating *ChR2-GFP* into the lateral ventricles of embryos at E14.5. The AAV-CAG-ChR2-GFP plasmid was a gift from Edward Boyden (Addgene #26929)[61]. One day after electroporation, the brains of embryos were removed and frozen on isopentane chilled with dry ice. The brain tissue was equilibrated in the cryostat for 2–3 h at −20 °C, then 20-μm-thick cryosections were collected on Superfrost Ultra Plus glass slides (ThermoFisher) and were held in the cryostat until finishing the sectioning. Sections were fixed with 10% ice-cold PFA for 30 min at 4 °C. After fixation, cryosections were washed with PBS and dehydrated with a series of alcohol solutions in ascending concentrations of 50% ethanol, 75% ethanol, and absolute ethanol each for 5 min, then slides were transferred to the last jar containing absolute ethanol and were incubated overnight at −20 °C. Next day, the RNAscope Multiplex Fluorescent Detection assay was performed based on manufacturer's protocol (Advanced Cell Diagnostics). The Protease IV treatment was shortened to 10 min to preserve enough GFP protein in the plasma membrane for sharp contour of individual cells in the VZ. The *Abhd4* RNAscope probe was custom designed (Mm-*Abhd4*-O1; #524551) and was used with Glast1 (Mm-*Slc1a3*-C2; #430781-C2) to visualize radial glia progenitor cells or intermediate progenitor cells, respectively. To enhance the membrane signal after RNAscope, sections were fixed with 10% PFA for 10 min at room temperature. After several, but brief rinsing steps with PBS, sections were incubated with goat antibody to green fluorescent protein (GFP, 1:1000, Abcam) or rabbit antibody to phospho-histone H3 (PHH3, 1:500, Millipore) and 5% NDS (Normal Donkey Serum, Sigma) solution at 4 °C overnight. Next day, slides were washed with PBS and treated in secondary antibody solution (Alexa Fluor 488-conjugated anti-goat or Alexa Fluor 647-conjugated anti-rabbit; 1:400, Jackson). The multiplex RNAscope signals were imaged by confocal microscopy in randomly selected GFP-expressing cells. The fluorescent dots representing individual mRNA molecules were quantified by manual counting for each marker and were normalized to the respective median value. To determine the segregation of *Abhd4* mRNA within the two parts of the dividing PHH3-positive cell pairs, *Abhd4* mRNA dots were counted manually within both compartments separately based on the mitotic cleavage furrow visualized by DAPI nuclear staining. The percentage of expression distribution between the two parts was calculated and the percentage values were plotted on the x axis histogram, with the *y* axis showing the number of cells with that percentage distribution. To quantify the total *Abhd4* mRNA dots number in the embryonic cortex, images were taken from three distinct layers: from the bottom of the VZ corresponding to VZ, from the cortical plate (CP) and from border of SVZ/IZ (intermediate zone). *Abhd4* mRNA quantity was determined by using an optimized ImageJ counter macro tool.

**Western blotting**. *Abhd4*-transfected HEK293 cells (as positive controls) and the telencephalon of wild-type and *Abhd4*-knock-out embryos (as negative controls) were homogenized in RIPA lysis buffer containing 50 mM Tris-HCl, 150 mM NaCl, 1% Triton X-100, 0.1% SDS, 1 mM DTT and 1X protease inhibitor cocktail (Roche). Cytosolic fractions of the samples were denatured in Laemmli sample buffer (Bio-Rad) for 5 min at 95 °C, and were loaded into a SDS-polyacrylamide gel (12%). Approximately 15 μg of total protein was separated in each lane at 160 V, 400 mA with PowerPac HC High-Current Power Supply and electrophoretically transferred to nitrocellulose membrane (Bio-Rad). To block non-specific binding in immunoblotting, 5% bovine serum albumin (Sigma) was used in Tris-buffered saline containing 0.1% Tween 20 buffer (TBST). The blots were incubated with the rabbit antibody to alpha/beta hydrolase 4 (ABHD4, 1:500, ImmunoGenes) in TBST overnight at 4 °C. After several washes with TBST, the membranes were treated with HRP-linked anti-rabbit secondary antibody (1:3000, Cell Signaling) for 2 h at room temperature and developed by SuperSignal West Dura Extended Duration Substrate Kit (ThermoFisher). Following read-out, the blots were stripped and incubated with the rabbit antibody to catalase (1:3000, Abcam) in TBST overnight at 4 °C, and treated as above to confirm comparable protein loading.

The primary antibody against the mouse ABHD4 was raised in transgenic rabbits that have elevated neonatal Fc receptor (FcRn) activity, because they carry one extra copy of the rabbit FcRn α-chain encoding gene (rabbit FCGRT)[62]. The rabbits (3-month-old females) were intramuscularly immunized with a keyhole limpet hemocyanin (KLH)-conjugated polypeptide (ABHD4: N′-PNQNKIWTVTVSPEQKDRT-C′) corresponding to amino acid residues 50–69 of the mouse ABHD4 enzyme. Immunization, antiserum selection, affinity purification and antibody validation in knockout mice was performed as described earlier[62]. All the treatments of rabbits in this research followed the guidelines of the Institutional Animal Care and Ethics Committee at ImmunoGenes Ltd that operated in accordance with permissions 22.1/601/000/2009 and XIV-I-001/2086-4/2012 issued by the Food Chain Safety and Animal Health Directorate of the Government Office of Pest County, Hungary.

**Fluorescence immunostaining and histology**. Embryonic brains were immersion fixed with 4% PFA overnight at 4 °C. Brains were cryoprotected in 15% sucrose in PBS buffer for 15 min and then in 30% sucrose solution overnight at 4 °C. After embedding into Tissue-Tek Optimal Cutting Temperature formulation (Sakura), the 20-μm-thick cryosections were collected on Superfrost Ultra Plus glass slides (ThermoFisher) and treated again with 4% PFA for 10 min. After several PBS washes, sections were permeabilized with 0.2% Triton X-100 in PBS and were blocked with 1% human serum albumin (Sigma) in PBS for one hour. Primary antibody incubation was carried out overnight at 4 °C. The following primary antibodies were used: rabbit antibody to phospho-histone H3 (PHH3, 1:500, Millipore), rabbit antibody to the transcription factor T-box, brain, 1 (TBR1, 1:500, Abcam), rabbit antibody to T-box, brain, 2 (TBR2, 1:500, Abcam), rabbit antibody to laminin subunit alpha 1 (LAMA1, 1:500, Sigma). In case of paired box protein-6 (PAX6)-immunostaining, sections were also treated after fixation with 10 mM citric acid for one hour at 65 °C, then washed in 0.1% Triton X-100 in PBS before the blocking step and the primary antibody incubation with rabbit antibody to PAX6 (1:300, Biolegend) and/or rat antibody to T-box, brain, 2 (TBR2, 1:300, Invitrogen). In case of BrdU labeling, the sections were treated with 2 M HCl for 1 h at 37 °C and then were blocked with 1% human serum albumin for 1 h before incubation with the mouse antibody to BrdU (1:1000, Sigma) overnight at 4 °C. The

following day, the sections were washed extensively in PBS buffer, and then incubated with the appropriate commercial Alexa Fluor 488-conjugated anti-mouse or anti-rabbit (1:400, Jackson), or Alexa Fluor 594-conjugated anti-rabbit (1:400, Jackson) secondary antibodies for 4 h at room temperature. To visualize F-actin-enriched adherens junctions in the ventricular wall, sections from the embryonic brains were treated as above, but antibody incubation was replaced by the high-affinity F-actin probe Alexa Fluor 568-Phalloidin (1:500, ThermoFisher) treatment for two hours at room temperature. To visualize cell nuclei, DAPI was included in the secondary antibody staining solution (1:1000 dilution of the 5 mg ml$^{-1}$ stock, Calbiochem). Cell death was detected by terminal deoxynucleotidyl transferase dUTP nick end labeling (TUNEL) assay by using Apoptag Red In Situ Apoptosis Detection Kit according to the manufacturer's protocol (Millipore). As the last step after each immunostaining and histological procedures, the slides were washed with PBS, mounted with Vectashield Hardset Antifade Mounting Medium (Vector), covered and sealed with nail polish.

In case of STORM super-resolution imaging, the embryonic brains were fixed in a similar manner, but were cut into thinner 20-μm-thick sections for better signal-to-noise ratio. Immunostaining was carried out in free-floating manner. Sections were permeabilized with 0.2% Triton X-100 in PBS and were blocked with 5% normal donkey serum (Sigma) in PBS for 1 h. Sections were incubated with a mouse antibody to nestin (1:200, Millipore) overnight at 4 °C, followed by intensive washing in PBS and then incubation with Alexa Fluor 647-conjugated anti-mouse (1:400, Jackson) secondary antibody for 4 h. Instead of using glass slides, the sections were dried onto coverslips, and stored uncovered at 4 °C until STORM imaging. Detailed information about the primary antibodies is listed in the Supplementary Information Table S2.

**In vitro experiments**. HEK293 cells were a kind gift from Balázs Gereben (Institute of Experimental Medicine). Cells were maintained under routine conditions in plastic Petri dishes in Dulbecco's Modified Eagle Medium (4.5 g l$^{-1}$ glucose, L-glutamine and sodium pyruvate; Corning) with 10% heat-inactivated fetal bovine serum (Biosera) in a 5% $CO_2$ atmosphere at 37 °C. Cells were seeded one day before transfection on poly-D-lysine-coated coverslips in 24-well culture plates. The cells were held in Opti-MEM Media (Gibco) for one hour, and then transfected with 1 μg plasmid DNA in Opti-MEM Media mixed with 2 μl Lipofectamine 2000 Reagent according to manufacturer's protocol (Invitrogen). After incubation for 20 h, the transfected cells were washed with PBS and then fixed with 4% PFA for 10 min. Subsequently, fixed cells were treated with 0.2% Triton X-100 for 15 min at room temperature and blocked with 1% human serum albumin in PBS for 30 min. The following primary antibodies were used for immunostaining: rabbit antibody to mitochondrial import receptor subunit TOM20 homolog (TOM20, 1:1000, Santa Cruz), mouse antibody to cytochrome c (CytC, 1:2000, Biolegend), and rabbit anti-cleaved-caspase 3 (CC3, 1:500; Cell Signaling). Primary antibodies were applied in PBS buffer for 1.5 h at room temperature. After washing steps, the cells were further incubated in CF568-conjugated anti-rabbit (1:1000, Biotium) secondary antibody and Alexa Fluor 647-conjugated anti-mouse (1:400 Jackson) secondary antibody in PBS buffer for 1 h at room temperature, and then rinsed extensively with PBS. In case of CC3-immunostaining, the coverslips were mounted with Vectashield Hardset Mounting Medium for confocal microscopy, whereas in case of the dual TOM20- and CytC-immunostaining, the sections were covered in imaging medium for STORM super-resolution microscopy.

**Microscopy**. Light micrographs were taken with an Eclipse 80i upright microscope equipped with a DS-Fi1 CCD camera (Nikon). High-resolution fluorescence images were obtained with an A1R confocal laser-scanning system built on a Ti-E inverted microscope and operated by NIS-Elements AR software (Nikon). STORM super-resolution images and the correlated high-power confocal stacks were acquired via a CFI Apo TIRF 100× objective (1.49 NA) on a Ti-E inverted microscope equipped with an N-STORM system, a C2 confocal scan head (Nikon), and an iXon Ultra 897 EMCCD camera (Andor)[63]. The HEK293 cells and the embryonic brain sections were covered with a freshly prepared imaging medium containing 0.1 M mercaptoethylamine and components of an oxygen scavenging system consisting of 5% ($m/v$) glucose, 1 mg ml$^{-1}$ glucose oxidase and 2.5 µl ml$^{-1}$ catalase in Dulbecco's PBS (Sigma). The coverslips were sealed with nail polish and transferred into the microscope setup after 10 min. In case of HEK293 cells, confocal $z$-stacks were taken from GFP-expressing cells and TOM20-positive mitochondria were selected to establish the best focal plane for STORM imaging. In case of brain sections, the GFP-positive radial processes of radial glia progenitor cells and the nestin-immunopositive intermediate filaments were located in the cortical plate in confocal z-stacks to determine the optical plane for subsequent STORM imaging. Continuous illumination with 405 nm laser line was used in order to reactivate the fluorophores and produce enough localization events, and 2500–5000 cycles were captured using 647-nm excitation. To acquire coordinates of localization points, peak detection was done using the Nikon N-STORM module in the NIS-Elements AR software.

**Image analysis**. To analyze correlated confocal and STORM images, pixel-based confocal images and 3D-coordinates of molecular localizations were loaded and aligned in the VividSTORM software[64]. To characterize the 3D-nanoarchitecture of nestin filaments, the number and density of localization points were calculated in randomly selected and identically sized filament segments as region-of-interests. After fitting a convex hull onto the outermost localization points, the volume and surface area of the filaments were also measured. Intensity profiles perpendicular to the nestin filaments were measured to calculate full-width-at-half-maximum values by fitting a Gaussian function in the NIS-Elements software. Gaussian rendering in the N-STORM module was used to visualize higher accuracy localizations by brighter dots. Convex hull and 3D-rendering images were obtained by the Visual Molecular Dynamics software[65]. To investigate mitochondria and the nanoscale distribution of cytochrome c in HEK293 cells, a freehand shape was first drawn around randomly selected GFP-expressing cells as region-of-interests. By using a custom Python script, the localization points representing cytochrome c were counted over the TOM20-positive and TOM20-negative pixels to establish cytochrome c distribution inside and outside of mitochondria, respectively.

Confocal microscopy was used to quantify cellular distribution, morphology, and cell death. In distribution analysis, the developing neocortex was divided into five laminar bins and the percentage of GFP-expressing electroporated cells was established in each bin by ImageJ. In morphological analysis, GFP-expressing cells were selected with ImageJ cell counter, rounded cells were counted manually and their percentage was determined. In cell death analysis, TUNEL-positive cell density was measured in the VZ/SVZ and in the CP with ImageJ cell counter. TUNEL$^+$/PAX6$^+$-positive cells were quantified manually. Cleaved caspase-3-positive and GFP-expressing cells were quantified with NIS-Elements Software.

The radial glia endfeet morphology was determined by manually using 15 µm thick confocal z-stack images. Branched morphology corresponds to multiple endfeet on one radial glia fiber, meanwhile club-like shape means one radial glia fiber with one endfeet[66].

The cell fate changes after $\Delta nCdh2$-GFP electroporation was measured using PAX6 and TBR2 double immunostaining. First electroporated cell number was quantified in both $\Delta nCdh2$-GFP and control conditions, then PAX6-positive, TBR2-positive, PAX6/TBR2 double-positive and double-negative cells were counted manually and their percentage distribution was calculated. Based on the findings reported in a recent study[67], the cell type-specific changes were determined by using the PHH3-immunostaining signal and cell morphology. bRG cells were counted as PHH3-positive cells connecting to the basal surface without apical connections. Apical radial glia cells were described as PHH3-positive cells along the ventricular surface, while IP cells were classified as PHH3-positive cells in the SVZ without any apparent surface connections.

The density of pyramidal cells, interneurons and astrocytes in the adult neocortex was quantified by Image J/Fiji. Images were auto-thresholded by the following functions: $vGlut1$-Otsu's; $Gad67$-Li's; $Glast1$-Triangle and converted to binary image. Pixel numbers with signal were determined in five equal bins and their percentages were counted.

Micrographs were always edited by equal settings between treatments, wild-type and $Abhd4$ knockout samples. Figure compositions were prepared by Photoshop CS5 (Adobe Systems, San Jose, CA).

**Prenatal ethanol exposure experiments**. Females heterozygous for the $Abhd4$ gene were bred with heterozygous males to ensure that littermate wild-type and $Abhd4$-knockout embryos are identically exposed to ethanol in utero. In the acute model, pregnant dams received a single intraperitoneal injection of either vehicle or 5 g kg$^{-1}$ ethanol in saline, and embryos (E14.5) were fixed 12 h later. In the subchronic model, vehicle or 2.5 g kg$^{-1}$ ethanol in saline was intraperitoneally injected twice a day between E13.5–E15.5, and embryos were collected at E16. Maternal blood ethanol content was determined enzymatically by using Synchron Systems Ethanol assay kit (Beckman). Blood ethanol standards were created (0.1; 0.5; 1; 1.5; 2‰). The dams' blood alcohol levels were between 0.5 and 1‰ in the subchronic model, whereas it reached 1.5–2‰ in the acute model.

**Phylogenetic tree**. Protein sequences from different phylogenetic levels were collected from UniProt (https://www.uniprot.org/) and NCBI (https://www.ncbi.nlm.nih.gov/) (accession numbers: Q8TB40, H2Q7Z0, Q8VD66, D3ZAW4, G1T725, Q5EA59, NP_001017287.1, NP_001017613.1, Q8WTS1, Q7JQU9, H2KZ86, A7S6S7). Sequence similarities were determined using protein alignment by Mega7 software[68]. The evolutionary history was inferred by the Maximum Parsimony method. The tree was obtained using the Subtree-Pruning-Regrafting algorithm incorporated into Mega7 interface.

**In silico single-cell RNAseq data**. In silico expression data were obtained from three different datasets from GEO database, accession links are provided in "Data Availability" section. Raw data was plotted without filtering.

**Statistical analysis and reproducibility**. Experimental results were tested for statistical significance using Statistica 13.1 (TIBCO) and Prism 5 (GraphPad). All treatments in each experiment were replicated at least in three independent cases.

Shapiro-Wilk normality test was used to measure normality of the samples. Unpaired comparisons were analyzed using two-sided Student's $t$ tests (normally distributed) and by Mann–Whitney $U$ tests (not normally distributed). Multiple comparisons were tested based on their parametric or nonparametric data by one-way ANOVA with post hoc Tukey's test or Kruskal–Wallis test followed by post hoc Dunn's test, respectively. Spearman's rank correlation coefficient was used to assess relationship between mRNA levels in the single cell-specific RNAscope experiments. The level of significance was set to $P < 0.05$ in every case, exact $P$ values are provided in each figure legend. Statistical graphs represent box-and-whisker plots including minima, maxima and mean or median values, and lower and upper quartiles) with single raw values or median values per animals. Precise details about the statistical graphs are provided in the figure legends and in the Source Data. Raw data and statistical information are reported in the Source Data. Experiments were replicated successfully at least in three independent cases and the number of replicates ($n$; both the number of sections and animals) is reported in the corresponding figure legends, except in case of qualitative comparisons. Micrographs in the figures are shown as representative images. No statistical methods were used to predetermine the sample size, it was defined based on the efficacy of electroporations in our laboratory. Samples were excluded from the data analysis due to compelling reasons, for example, failed experiments or unsuccessful electroporation.

**Reporting summary**. Further information on research design is available in the Nature Research Reporting Summary linked to this article.

## Data availability

The data that support the findings of this study are available from the corresponding author upon reasonable request. Source data are provided with this paper which includes all statistical results regarding Figs. 1k, p, 2k, t, 3k, n, 4c, f, o, p, 5d, h, o, 6e, l, m, 7i and Supplementary Figs. 1m, n, 2j, q, 3d, 5j, k, 6c, f, i, 7c, f, i, 9k, 10g, h, i, j, q, r, 11e, 12i, l as well as uncropped images of western blots. In silico analysis were made by using two different datasets: GSE38805 (embryonic mouse cortex) and GSE75140 (fetal human cortex and cerebral organoid). Source data are provided with this paper.

## Code availability

Custom scripts and software code generated for supporting the findings of this study are available on request from the corresponding author.

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

## Acknowledgements

The authors thank Dr. E. Horváth, B. Pintér, E. Tischler for laboratory support, Drs. G. Balogh, Z. Balogi, A. Dorning, B. Dudok, M. Melis, M. Péter, S. Prokop, L. Vígh for help with preliminary experiments and comments. The authors appreciate the help of Dr. Ozge Gunduz-Cinar and Dr. Andrew Holmes with the RNAscope protocol. The authors are grateful to the IEM Medical Genetics Unit for mouse colony management, to S.S.J. Hu, K. Nagy for antibody generation, to Dr. B. Gereben for providing HEK293 cells and to K. Balogh for artwork. The help of Dr. L. Barna, the Nikon Microscopy Center at the Institute of Experimental Medicine, Nikon Europe B.V., Nikon Austria GmbH and Auro-Science Consulting is acknowledged for kindly providing microscopy support. This work was supported by the National Brain Research Program (2017-1.2.1-NKP-2017-00002), by the National Research, Development and Innovation Office, Hungary (VKSZ-14-1-2O15-0155 for antibody generation; VEKOP-2.3.3-15-2016-00013 for super-resolution microscopy development; K116915 for ABHD4 research and Frontier Program 129961 for cannabinoid research). I.K. holds the Naus Family Chair in Addiction Sciences in the Department of Psychological and Brain Sciences at Indiana University Bloomington and his work is also supported by the National Institutes of Health (R01NS099457 and R01DA044925). Additional support was provided by the Semmelweis University Grant (EFOP-3.6.3-VEKOP-16-2017-00009 to Z.I.L.), the US National Institutes of Health Grant (DA021696 and DA011322) to K.M. and (DA037660) to B.F.C.

## Author contributions

I.K., Z.L., and K.M. conceived the project and designed the experiments. Z.I.L. and Z.L. carried out molecular cloning, in situ hybridization, immunostaining, confocal microscopy and data analysis. Z.I.L. carried out western blots, phylogenetic analysis and the in vivo experiments. Z.I.L. and F.M. performed the in vitro experiments. M.Z., V.M., and F.M made STORM imaging and data analysis. G.M.S and B.F.C provided the *Abhd4*-knockout mouse line, I. Kacskovics and K.M. developed and Z.I.L. validated ABHD4 antibodies. Z.L. and I.K. wrote the manuscript with the help of all co-authors.

## Competing interests

I. Kacskovics is a scientific co-founder of ImmunoGenes Ltd., a company specialized in the generation of FcRn transgenic animals for the production of polyclonal and monoclonal antibodies. All the other authors declare no competing interests.

## Additional information



