## [Peer Review File · Nature Communications]

Reviewers' Comments:

Reviewer #1:

Remarks to the Author:

Prof. Katona and colleagues proposed developmental anoikis and the molecular player that controls this process. Developmental anoikis was triggered as a result of exposure to different types of insults. One is forced delamination of neural progenitor cells by overexpression of Cdh2 mutant. Another is prenatal alcohol exposure. Concept of "developmental anoikis" that reduces errors of development that is elicited by environmental insults is novel. Based on the known function in cancer anoikis, and the specific expression in developmental stage, the authors picked up Abhd4 as a candidate gene of a controller of this event. The knockout animal did not show obvious phenotypes in normal condition. However, under exposure to insults, the anoikis is significantly reduced in the knockout. The authors also provide good discussion of potential mechanisms how Abhd4 controls developmental anoikis. This article provides very interesting model, and I believe the model will be further strengthened by answering the questions below.

Abhd4 expression is remarkably specific to radial glial progenitor cells. Did authors observe reduction of cell death only in that cell population (for example, no changes in cortical plate?) in the knockout cortex of Abhd4 in cases of delta cdh2 overexpression and FASD model? A related comment; is Abhd4 expression altered by prenatal alcohol exposure? For example, do cells other than radial glial progenitor cells increase the expression of Abhd4, so that reduction of cell death is not limited to radial glial progenitor cells in the knockout? Depending on answers to those questions, proposed model may also require revision.

The effects on migration and cell death are obvious in embryonic stages in Abhd4 manipulation. Are those effects kept until adulthood and make abnormal adult brain structure, or do cells catch up migration? This information is important for discussion about significance of developmental anoikis.

Other comments:

Most of TUNEL labelling did not overlap with GFP. Does cell death happen in cell non autonomous manner?

Reviewer #2:

Remarks to the Author:

With this paper the Katona's lab seeks to introduce a new concept in the field of developmental neuroscience: the developmental anoikis as a safeguarding mechanism responsible for the elimination of abnormally delaminated apical progenitors (RGPCs) during brain development. The authors identified (i) ABDH4 as a crucial molecular player in developmental anoikis and (ii) developmental anoikis as one of the possible mechanisms responsible for the elimination of abnormally delaminated cells in a pathological condition, fetal alcohol syndrome.

The paper is a pleasant read and the ideas illustrated by the authors bring a fresh perspective in the neurogenesis field. I acknowledge that the authors address an issue that is under-represented in our field: how robustness is achieved in the context of cortical development and brain morphogenesis.

Given the potential of their work, the authors must address the following points to make their paper suitable for acceptance and publication in Nature Communication.

MINOR POINTS

- A general remark. Throughout the text, the authors refer to abnormal RGPC delamination and

AJs disassembly. Strictly speaking, delamination is always paralleled by the cell's disengagement from the AJs belt. What abnormal exactly means? What is different between a normal and an abnormal AJ disassembly/delamination? What is then setting the difference?

The same apply to the use of the word "premature", referred to delamination. Premature compared to what exactly? Do the authors mean a delamination that is not paralleled by a fate transition? Or a delamination that is not sync with cell cycle progression? The authors must clarify and elaborate conceptually on that.

- Page 5: the authors write "spatially limited destruction..." upon nCadh20-GFP manipulation referring to Figure1 a-d.

However, the panels show a deep and generalized modification of the AJs. The authors must clarify this point and make clear if the image is representative of their conclusions. Is the manipulation really causing a limited change in the AJ?

If so, then the authors should choose a picture reflecting this claim.

Otherwise, the authors should change the sentence to reflect what the picture showed: a pronounced and general change in the AJs.

The same apply to the phalloidin staining shown in Supplementary Figure 1a-f. The phalloidin staining at the apical surface is clearly less intense in the nCadh20-GFP -electroporated area compared to control electroporation area. This is of course compatible with a junctional perturbation, but it suggests that the perturbation is generalized, and not is "spatially limited".

- Regarding the effect of the manipulation on the basal endfeet. I believe this data is crucial in the clarification of the specificity of the cell biological effects of the experimental manipulation. The authors must show the data now parked in Supplementary Figure 1j-k in the main figure, possibly in Figure 1.

Furthermore, given previous literature on the dynamics of the basal endfeet (Yokota et al, 2010), it would be important to define if any morphological change happens at the basal endfoot. I am not asking the author to generate new data, but rather to show the data they already have in a quantitative manner. For example, the authors should show a classification/quantification of the structure of the basal end feet as in Yokota et al, 2010, Figure 2 (club like vs branched end feet). In addition, the authors should add a quantification of the data shown in Supplementary Figure 1m-n.

- Page 5. The delaminated progenitors generated upon nCadh20-GFP retain the basal process, Pax6 expression and show an increased apoptosis

(i) which is the % of delaminated cells that are also apoptotic?

(ii) For those cells that are not apoptotic: do these delaminated cells correspond to bRG/oRG cells?

The authors must address this point.

I would suggest doing 3D reconstruction of their STORM data (or other suitable data sets) to highlight the complete morphology of the manipulated cells in the SVZ. Additionally, I recommend a staining with p-Vim to understand if there are more cells in SVZ retaining the basal process in mitosis, as hallmark of bRGs.

- As for scRNAseq public libraries: it would be informative for the readers to show the actual data/plots for the ABDH4 expression in different cell types in the mentioned libraries. These data can be added as or in a supplementary figure.

- Based on the data shown by the authors, the main role of ABDH4 seems to be associated with its downregulation. Based in the author's claim, ABDH4 downregulation takes place immediately after delamination. This point is interesting and it is crucial in defining the mechanism of action of ABDH4. However, I think the claim is not fully supported by the data. It would be very informative to show an immunostaining to define the localization of ABDH4. I am actually surprised that, given their experience with high resolution microscopy, the authors limit their attention the mRNAs, without investigating the protein localization. Please clarify if this is technical issue related to the Ab availability. Is ABDH4 -both at the mRNA and protein level- associated with the AJs or to

the asymmetric partitioning of ABDH4 during RGPC mitosis? That could provide a cellular mechanism for the fast downregulation of ABDH4 in delaminated cells. Investigating this point would bring the entire paper to a higher level.

- Page 7-8: the observation of the inverse correlation of ABDH4 with Tbr2 mRNA is interesting. However, I urge the authors to consider their data in light of previously published work. Of note, it has been demonstrated that aRGs cells already express Tbr2 mRNA (see Florio et al, 2015). So, Tbr2 mRNA is not a specific marker for delaminated IPs, as it is present also in not a delaminated IPs, and more importantly also in aRG cells (see Wilsch-Brauninger et al, 2012 and Florio et al, 2015). In addition, the authors mentioned that they consider cells in the VZ. Again, the VZ localization is not indicative of a certain fate, as VZ contain APs, and IPs, both non-delaminated and delaminated. It is not clear if the cells analyzed by the authors are retaining or not the apical contact. I think the authors should run an additional quantification in which they score separately cells with and without the apical contact. That will help with the interpretation of the graphs shown in Figure panel X, that in its present state is hard to interpret and not particularly convincing.

MINOR POINTS

- Page 65-66: the model should be added as last panel to the main figure.

- Page 3: the word INTRODUCTION is missing.

- the paper address the issue of how robustness is achieved in the context of cortical development and brain morphogenesis. I would find extremely interesting if the authors could elaborate more on that in the discussion part.

Reviewer #1:

- Prof. Katona and colleagues proposed developmental anoikis and the molecular player that controls this process. Developmental anoikis was triggered as a result of exposure to different types of insults. One is forced delamination of neural progenitor cells by overexpression of Cdh2 mutant. Another is prenatal alcohol exposure. Concept of “developmental anoikis” that reduces errors of development that is elicited by environmental insults is novel. Based on the known function in cancer anoikis, and the specific expression in developmental stage, the authors picked up Abhd4 as a candidate gene of a controller of this event. The knockout animal did not show obvious phenotypes in normal condition. However, under exposure to insults, the anoikis is significantly reduced in the knockout. The authors also provide good discussion of potential mechanisms how Abhd4 controls developmental anoikis. This article provides very interesting model, and I believe the model will be further strengthened by answering the questions below.

We would like to thank the Reviewer for his/her supportive comments and for highlighting that the concept of developmental anoikis is novel. We are also grateful for his/her excellent recommendations on how to strengthen our study. We share his/her view that the suggested new experiments were important, and the new data substantially consolidated our original conclusion. We look forward to learning his/her opinion about the revised version of the manuscript.

- Abhd4 expression is remarkably specific to radial glial progenitor cells. Did authors observe reduction of cell death only in that cell population (for example, no changes in cortical plate?) in the knockout cortex of Abhd4 in cases of delta cdh2 overexpression and FASD model? A related comment; is Abhd4 expression altered by prenatal alcohol exposure? For example, do cells other than radial glial progenitor cells increase the expression of Abhd4, so that reduction of cell death is not limited to radial glial progenitor cells in the knockout? Depending on answers to those questions, proposed model may also require revision.

We fully agree with the Reviewer about the importance of these questions. To this end, we have re-analyzed the TUNEL stainings in the developing cerebral cortex of wild-type and littermate *Abhd4*-knockout mice after *in utero* electroporation of a dominant-negative version of N-cadherin (*ΔnCdherin-GFP*) and after prenatal ethanol exposure. The new results are described on page 13 and 14 and are presented in Fig. 6m and Fig. 7i. Importantly and in accordance with our original model, we did not observe an increase in cell death levels in the cortical plate indicating that differentiated neurons are not affected. Additionally, acute maternal ethanol treatment did not increase *Abhd4* levels in the cortical plate of the embryos (new Supplementary Fig. 12j-l). These two findings together further corroborate the notion that ABHD4-dependent cell death is restricted to the proliferative zones of the developing cerebral cortex.

- The effects on migration and cell death are obvious in embryonic stages in *Abhd4* manipulation. Are those effects kept until adulthood and make abnormal adult brain structure, or do cells catch up migration? This information is important for discussion about significance of developmental anoikis.

We thank the Reviewer for this very important question. In order to address this issue, we carried out short-term and long-term survival experiments (see on page 11 and in new Supplementary Fig. 9). By using co-electroporation of red and the *Abhd4*-linked green indicators (tdTomato and GFP, respectively)-containing plasmids, the tdTomato and GFP marker proteins colocalized 24 hours after electroporation (new Supplementary Fig. 9a-f). Next, we examined the fate of the electroporated cells at postnatal day 3 (P3), by which time point all healthy postmitotic neurons that could escape cell death should be able to reach the cortical plate. We noticed that a proportion of electroporated cells could catch up with migration by P3 as suggested by the Reviewer (new Supplementary Fig. 9g,j) indicating the on-going interaction of pro-survival and pro-apoptotic processes in delaminated cells. To determine whether specific elimination of the pro-apoptotic overexpressed ABHD4 renders postmitotic neurons capable to survive and enables their delayed migration, we compared the effects of a fusion protein construct (*Abhd4-GFP*) and a bicistronic (*Abhd4-IRES-GFP*) construct. By using the fusion protein construct, we only observed tdTomato-expressing pyramidal neurons in layer 2/3 that were completely devoid of the GFP signal (new Supplementary Fig. 9g,h). This demonstrates that only those neurons could reach the cortical plate that are able to fully downregulate the ABHD4-GFP fusion protein. In striking contrast, when we co-electroporated the bicistronic construct in which the transcription of *Abhd4* and *GFP* is coupled, but translation of the two proteins is independent due to the internal ribosomal entry site, we found that all tdTomato-expressing pyramidal neurons that survived the effects of *Abhd4* overexpression and could migrate into the cortical plate are also GFP-positive (new Supplementary Fig. 9i,j). This finding suggests that postmitotic neuroblasts specifically degrade ABHD4, but leave tdTomato and GFP proteins intact, and then undergo delayed migration. In accordance with the sharp downregulation of endogenous *Abhd4* mRNA expression in the committed daughter cells (Figure 3), this new observation indicates that ABHD4 protein levels are also kept under tight control by intra- and/or extracellular signals in postmitotic neuroblasts. Although it is beyond the scope of the present study, our intention is to pursue these regulatory mechanisms in the future.

Other comments:

- Most of TUNEL labelling did not overlap with GFP. Does cell death happen in cell non autonomous manner?

The Reviewer correctly noticed the relatively low colocalization ratio between GFP and TUNEL labeling. Because the TUNEL method (Terminal deoxynucleotidyl transferase dUTP Nick End Labeling) detects the DNA breaks formed when DNA fragmentation occurs in the last phase of apoptosis, it is conceivable that the majority of the GFP protein has already become degraded and hence GFP visualization does not reach the detection threshold within dying cells 48 hours after the electroporation, when most TUNEL-positive cells have already displayed disintegrated structure. We are grateful to the Reviewer for raising the importance to clarify this issue. Thus, we have also performed new experiments to investigate cell death 24 hours after electroporation (described on page 6 and shown in new Fig. 2a-k) when cell integrity may still be better preserved. We found that GFP and TUNEL signals significantly overlapped at this earlier time point. Notably, GFP distribution was already restricted to small compartments within the cells indicating its on-going degradation (see some examples in Fig. 2c). We have also exploited the better retained immunogenicity of dying cells at this earlier time point in order to reply also to the question of Reviewer 2 (see his/her 4th comment below). We have established that at least 73% of TUNEL-positive cells also showed PAX6-immunostaining when counting TUNEL-positivity in those cells that still retained their normal cellular structure. As an important consideration, this ratio is a minimum estimation and false immunonegativity due to degraded PAX6 proteins in cells with more progressed stage of apoptosis cannot be excluded.

Nevertheless, the high colocalization of TUNEL-labeling with GFP and PAX6 together indicate that directly affected radial glia progenitor cells are dying predominantly in a cell-autonomous manner in agreement with our original model. In addition, the *in vitro* experiments in the HEK cell assay further support the scenario that the signaling mechanism in ABHD4-dependent cell death is primarily cell-autonomous (described on page 12 and shown in Supplementary Fig. 10k-r). Certainly, it is also plausible to assume that some non-electroporated cells that are largely surrounded by electroporated neighbours are also pathologically detached due to the loss of their cell-cell anchors with their affected neighbours. In this sense, the cause of cell death for some cells can indeed be considered as non-cell-autonomous, although we would rather refer to this as an indirect effect.

Reviewer #2:

- With this paper the Katona's lab seeks to introduce a new concept in the field of developmental neuroscience: the developmental anoikis as a safeguarding mechanism responsible for the elimination of abnormally delaminated apical progenitors (RGPCs) during brain development. The authors identified (i) ABDH4 as a crucial molecular player in developmental anoikis and (ii) developmental anoikis as one of the possible mechanisms responsible for the elimination of abnormally delaminated cells in a pathological condition, fetal alcohol syndrome.

The paper is a pleasant read and the ideas illustrated by the authors bring a fresh perspective in the neurogenesis field. I acknowledge that the authors address an issue that is under-represented in our field: how robustness is achieved in the context of cortical development and brain morphogenesis.

We appreciate very much the encouraging comments of the Reviewer as well as his/her helpful and constructive criticisms, which helped us to improve several technical and conceptual aspects of the study. We have addressed all points raised either with new experiments, measurements or analyses and by rewriting the relevant parts of the manuscript to improve clarity.

- A general remark. Throughout the text, the authors refer to abnormal RGPC delamination and AJs disassembly. Strictly speaking, delamination is always paralleled by the cell's disengagement from the AJs belt. What abnormal exactly means? What is different between a normal and an abnormal AJ disassembly/delamination? What is then setting the difference?

The same apply to the use of the word "premature", referred to delamination. Premature compared to what exactly? Do the authors mean a delamination that is not paralleled by a fate transition? Or a delamination that is not sync with cell cycle progression? The authors must clarify and elaborate conceptually on that.

Thank you for this important comment. As the Reviewer correctly suggests, our wording ("abnormal" and "premature") was intended to illustrate that forced disassembly of adherens junctions results in misplaced radial glia progenitor cells that are not supposed to delaminate under control conditions. Because the previously used terms "abnormal" and "premature" are indeed ambiguous, we replaced these terms and use "delamination" and "pathological detachment" to refer to control or perturbed conditions, respectively, throughout the revised text.

- Page 5: the authors write “spatially limited destruction...” upon nCadh20-GFP manipulation referring to Figure 1 a-d.

However, the panels show a deep and generalized modification of the AJs. The authors must clarify this point and make clear if the image is representative of their conclusions. Is the manipulation really causing a limited change in the AJ?

If so, then the authors should choose a picture reflecting this claim.

Otherwise, the authors should change the sentence to reflect what the picture showed: a pronounced and general change in the AJs.

The same apply to the phalloidin staining shown in Supplementary Figure 1a-f. The phalloidin staining at the apical surface is clearly less intense in the nCadh20-GFP - electroporated area compared to control electroporation area. This is of course compatible with a junctional perturbation, but it suggests that the perturbation is generalized, and not is “spatially limited”.

We are grateful to the Reviewer for calling our attention to the fact that the high magnification micrograph does not illustrate well enough the spatially limited destruction of adherens junction. When using the words „spatially limited” or restricted, we refer to the fact that AJ disruption closely follows the ectopic expression of $\Delta nCdh2$ -GFP. Under no circumstances do we intend to state that the AJ is being broken down only partially in the electroporated area. In order to make this point clear for the readers, we altered the manuscript text and edited the low power image in Supplementary Fig. 1d-f to better illustrate that disruption of the adherens junction belt marked by phalloidin is restricted to the same area that contains radial glia progenitor cells affected by the ectopic expression of $\Delta nCdh2$ -GFP.

- Regarding the effect of the manipulation on the basal endfeet. I believe this data is crucial in the clarification of the specificity of the cell biological effects of the experimental manipulation. The authors must show the data now parked in Supplementary Figure 1j-k in the main figure, possibly in Figure 1.

Furthermore, given previous literature on the dynamics of the basal endfeet (Yokota et al, 2010), it would be important to define if any morphological change happens at the basal endfoot. I am not asking the author to generate new data, but rather to show the data they already have in a quantitative manner. For example, the authors should show a classification/quantification of the structure of the basal end feet as in Yokota et al, 2010, Figure 2 (club like vs branched end feet).

In addition, the authors should add a quantification of the data shown in Supplementary Figure 1m-n.

We thank the Reviewer for these important suggestions. As recommended, we moved the representative confocal microscopy images to main Figure 1e-j. In addition, we quantified the structure of the basal end feet as in Yokota et al 2010 and found no difference in the ratio of the morphological types. This observation further corroborates the specific impact of $\Delta nCdh2$ -GFP *in utero* electroporation on the apical adherens junctions. The new data are described on page 5 and are shown in main Fig. 1k. We also quantified the STORM super-resolution imaging data on the nanoarchitecture of the radial glia scaffold as requested and added the new data in Supplementary Fig. 1m,n.

- Page 5. The delaminated progenitors generated upon nCadh20-GFP retain the basal process, Pax6 expression and show an increased apoptosis which is the % of delaminated cells that are also apoptotic?

For those cells that are not apoptotic: do these delaminated cells correspond to bRG/oRG cells? The authors must address this point.

I would suggest doing 3D reconstruction of their STORM data (or other suitable data sets) to highlight the complete morphology of the manipulated cells in the SVZ. Additionally, I recommend a staining with p-Vim to understand if there are more cells in SVZ retaining the basal process in mitosis, as hallmark of bRGs.

We have carried out new experiments and data analysis to address these interrelated questions and suggestions. Precise quantification of all delaminated cells undergoing apoptosis was not possible for two reasons. At any given time point only a selected population of cells reach the TUNEL-positive late stage of apoptosis. The TUNEL signal visualizes DNA breaks hence by the time it appears many cells may have already lost their cellular integrity and immunogenicity for marker proteins such as PAX6. However, to experimentally address this important issue and provide at least some estimates, we carried out triple TUNEL-, GFP- and PAX6-staining to measure the approximate proportion of radial glia progenitor cells undergoing cell death. As summarized on page 6 and shown in new Fig. 2e-k, when counting all TUNEL-positive signals we found that 30% of these signals still retained PAX6-immunopositivity. Lots of these TUNEL signals however already lost their cellular integrity suggesting that they might have also lost Pax6-immunogenicity in parallel. Notably, we found that the majority (~73%) of TUNEL-labeled cells with intact morphology were still PAX6-positive 24 hours after electroporation (see also Reply to Reviewer 1, third comment).

To determine if some of the delaminated cells correspond to basal/outer radial glia (bRG/oRG) cells, we performed a new set of *AnCdh2-GFP* electroporation experiments. Unfortunately, the limited z-depth (600 nm) available in our STORM microscope setup does not allow the complete reconstruction of a radial glia progenitor cell. Therefore, we used confocal microscopy to analyze both cell-fate and morphology in the new experiment. Because p-Vimentin-immunostaining did not work reliably despite trying several p-Vimentin antibodies, we used the mitotic marker PHH3 based on the recent findings of Namba et al. 2020, Neuron, to visualize the dividing cells. Based on the presence of the basal process in mitotic cells in the subventricular zone, we found an increased percentage of basal radial glia bRG cells (and also intermediate progenitor cells) at the expense of aRGs. In addition, by using PAX6- and TBR2-immunostaining, we observed a conversion of cell fate in agreement with the previous findings of Zhang et al, 2010, Developmental Cell. The new results are detailed on page 5-6, and presented in new Supplementary Fig. 2.

- As for scRNAseq public libraries: it would be informative for the readers to show the actual data/plots for the *ABHD4* expression in different cell types in the mentioned libraries. These data can be added as or in a supplementary figure.

As requested, we present *Abhd4* expression data together with *Pax6* and *Tbr1* mRNA expression obtained from three publicly available representative scRNA-seq datasets (embryonic mouse cortex from Fietz et al. 2012; fetal human neocortex and cerebral human organoid from Camp et al 2015) on page 7 and in new Supplementary Fig. 4. The data show that *Abhd4* levels mirror high *Pax6* mRNA expression in the ventricular zone, but its expression is very low in differentiated cortical plate cells characterized by high *Tbr1* levels.

- Based on the data shown by the authors, the main role of ABDH4 seems to be associated with its downregulation. Based in the author's claim, ABDH4 downregulation takes place immediately after delamination. This point is interesting and it is crucial in defining the mechanism of action of ABDH4. However, I think the claim is not fully

supported by the data. It would be very informative to show an immunostaining to define the localization of ABDH4. I am actually surprised that, given their experience with high resolution microscopy, the authors limit their attention to the mRNAs, without investigating the protein localization. Please clarify if this is a technical issue related to the Ab availability. Is ABDH4 -both at the mRNA and protein level- associated with the AJs or to the asymmetric partitioning of ABDH4 during RGPC mitosis? That could provide a cellular mechanism for the fast downregulation of ABDH4 in delaminated cells.

We fully agree with the Reviewer, and indeed, we have been trying hard to obtain an antibody that recognizes the ABHD4 protein in fixed embryonic cortical tissue preparations. We have tested 10 independent commercial and custom-made antibodies by using antigens covering the entire length of the protein during the course of this project. One of our custom-made antibodies worked relatively well for Western-blotting under denaturing conditions as demonstrated by control experiments on *Abhd4*-knockout embryonic cortical tissue. However, only non-specific background labeling was achieved in fixed tissue preparations despite our efforts with several antigen retrieval methods. This could be due to the masking of the potential antigen sites by the tertiary structure of the protein or by ABHD4-binding proteins. Alas, we were restricted to analyzing mRNA distribution in our experiments. Nevertheless, we completely agree with the Reviewer that a potential asymmetric segregation of ABHD4 is possible and would be highly interesting. Therefore, we have carried out experiments to test the possibility whether *Abhd4* mRNA is differentially segregated during mitosis. We describe the results on page 8-9 and present the data in new Fig. 3I-n. By using PHH3-immunostaining, we have selected mitotic pairs of cells that are still attached and analyzed the percentage distribution of *Abhd4* expression between the daughter cells with RNAscope *in situ* hybridization. The analysis revealed no difference in the expression levels as demonstrated by the single-peak Gaussian distribution of the plot.

- Page 7-8: the observation of the inverse correlation of ABHD4 with *Tbr2* mRNA is interesting. However, I urge the authors to consider their data in light of previously published work. Of note, it has been demonstrated that aRGs cells already express *Tbr2* mRNA (see Florio et al, 2015). So, *Tbr2* mRNA is not a specific marker for delaminated IPs, as it is present also in not a delaminated IPs, and more importantly also in aRG cells (see Wilsch-Brauninger et al, 2012 and Florio et al, 2015). In addition, the authors mentioned that they consider cells in the VZ. Again, the VZ localization is not indicative of a certain fate, as VZ contain APs, and IPs, both non-delaminated and delaminated. It is not clear if the cells analyzed by the authors are retaining or not the apical contact. I think the authors should run an additional quantification in which they score separately cells with and without the apical contact. That will help with the interpretation of the graphs shown in Figure panel X, that in its present state is hard to interpret and not particularly convincing.

Thank you for this important comment. As requested, we removed the *Tbr2-Abhd4* mRNA expression correlation plot, because *Tbr2* mRNA is also present in radial glia cells. Unfortunately, the lack of a proper antibody against ABHD4 deprived us from doing the simple double immunohistochemistry experiment which could have simply answered the question. To circumvent this issue, we have carried out *in situ* hybridization to visualize *Abhd4* mRNA in combination with TBR2-immunostaining. This experiment clearly revealed the spatial

segregation of *Abhd4* mRNA-expressing cells from the TBR2 protein-containing cells. This experiment and the new data are presented on page 9 and in new Fig. 3o-q.

MINOR POINTS

- Page 65-66: the model should be added as last panel to the main figure.

We are very glad that the Reviewer judged the ABHD4-dependent developmental anoikis model and its schematic representation worth to be presented in a main figure. Accordingly, we added this figure as a main figure (Fig. 8).

- Page 3: the word INTRODUCTION is missing.

We thank the Reviewer for noticing this omission and added “Introduction”.

- The paper address the issue of how robustness is achieved in the context of cortical development and brain morphogenesis. I would find extremely interesting if the authors could elaborate more on that in the discussion part.

We were delighted to learn that the Reviewer shares our view that developmental anoikis has strong significance for the concept of robustness in the developing brain. We have added a few sentences to the Discussion on page 17-18 to further emphasize the importance of developmental anoikis in the context of robustness and in developmental biology.

Reviewers' Comments:

Reviewer #1:

Remarks to the Author:

The authors addressed all of points and the manuscript reads well.

Reviewer #2:

Remarks to the Author:

The revised version of László et al. addresses the points raised during the previous round of review.

I appreciate the fact that in the present version the authors provide additional data and quantifications corroborating and strengthening their original claims.

It is my opinion that the paper in its present form should be accepted for publication.

Only one point should be addressed: there are here and there several typos, that must be corrected and edited before publication.

Reviewer #1 (Remarks to the Author):

- The authors addressed all of the points and the manuscript reads well.

We thank the Reviewer's supportive comments and we are glad that we could answer all important questions.

Reviewer #2 (Remarks to the Author):

- The revised version of Laszlo et al. addresses the points raised during the previous round of review. I appreciate the fact that in the present version the authors provide additional data and quantifications corroborating and strengthening their original claims.

It is my opinion that the paper in its present form should be accepted for publication. Only one point should be addressed: there are here and there several typos, that must be corrected and edited before publication.

We are very grateful for the comments of the Reviewer as well as his/her support for the publication of our manuscript. Also, we would like to thank the Reviewer's constructive and helpful comments and questions raised in the previous round and we are glad that we were able to answer all of these important points which improved our manuscript. We also appreciate for calling our attention to the typos, we performed a scrutinized spell check and corrected them.